



# Potential environmental impact of bromoform from Asparagopsis farming in Australia

Yue Jia[1, a], Birgit Quack[2*], Robert D. Kinley[3], Ignacio Pisso[4], Susann Tegtmeier[1]

[1] Institute of Space and Atmospheric Studies, University of Saskatchewan, Saskatoon, Canada

[2] GEOMAR Helmholtz Centre for Ocean Research Kiel, Kiel, Germany

[3] Commonwealth Scientific and Industrial Research Organisation (CSIRO), Agriculture and Food, Townsville, QLD 4811, Australia

[4] Norwegian Institute for Air Research (NILU), Kjeller, Norway

[a] *now at*: Cooperative Institute for Research in Environmental Sciences (CIRES), University of Colorado Boulder, Boulder, CO, USA.

*Corresponding to*: B. Quack (bquack@geomar.de)





**Abstract**
To mitigate the rumen enteric methane ($CH_4$) produced by ruminant livestock, *Asparagopsis*
*taxiformis* is proposed as an additive to ruminant feed. During the cultivation of *Asparagopsis*
*taxiformis* in the sea or in terrestrial based systems, this macroalgae, like most seaweeds and
phytoplankton, produces a large amount of bromoform ($CHBr_3$), which may contribute to ozone
depletion once released into the atmosphere. In this study, the impact of $CHBr_3$ on the stratospheric
ozone layer resulting from potential emissions from proposed *Asparagopsis* cultivation in
Australia is assessed by weighting the emissions of $CHBr_3$ with the ozone depletion potential
(ODP), which is traditionally defined for long-lived halogens but has been also applied to very
short lived substances (VSLSs). An annual yield of ~$3.5 \times 10^4$ Mg dry weight (DW) is required
to meet the needs of 50% of the beef feedlot and dairy cattle in Australia. Our study shows that the
intensity and impact of $CHBr_3$ emissions varies dependent on location and cultivation scenarios.
Of the proposed locations, tropical farms near the Darwin region are associated
with largest $CHBr_3$ ODP values. However, farming of Asparagopsis using either ocean or
terrestrial cultivation systems at any of the proposed locations does not have potential to impact
the ozone layer. Even if all *Asparagopsis* farming was performed in Darwin, the emitted $CHBr_3$
would amount to less than 0.016% of the global ODP-weighted emissions. The remains are
relatively small even if the intended annual yield in Darwin is scaled by a factor 30 to meet
the global requirements, which will increase the global ODP-weighted emissions by 0.48%




## 1. Introduction

Livestock is responsible for about 15% total anthropogenic greenhouse gas (GHG) emissions (Gerber et al., 2013), ranking it amongst the main contributors to climate change. The global demand for red meat and dairy is expected to increase >50% by 2050 compared to 2010 level, thus mitigation measures to reduce the GHG emission from the global livestock industry are in high demand (Beauchemin et al., 2020). Total GHG emissions from ruminant livestock contribute about 18% of the total global carbon dioxide equivalent ($CO_2$-eq) inventory as $CH_4$ (Herrero and Thornton, 2013). With a global warming potential 28 times higher than carbon dioxide ($CO_2$) and a much shorter lifetime (~10 years, IPCC, 2014), ruminant enteric $CH_4$ is an attractive and feasible target for global warming mitigation.

Enteric $CH_4$ from ruminant livestock is produced and released into the atmosphere through rumen microbial methanogenesis (Morgavi et al., 2010). Methanogenic archaea (methanogens) intercept substrate $CO_2$ and $H_2$ liberated during bacterial fermentation of feed materials (Kamra, 2005), and during this inefficiency of the digestion process (Herrero and Thornton, 2013; Patra, 2012), methanogen metabolism leads to reductive $CH_4$ production and loss of feed energy as $CH_4$ emissions. To abate enteric methanogenesis, different strategies such as feeding management and antimethanogenic feed ingredients, have been proposed and assessed (e.g., Moate et al., 2016; Mayberry et al., 2019; Beauchemin et al., 2020). Some types of macroalgae have been demonstrated to mitigate production of $CH_4$ during *in vitro* and *in vivo* rumen fermentation significantly (Kinley and Fredeen 2015; Li et al., 2018; Kinley et al., 2020; Abbott et al., 2020). Among the different macroalgae species, Kinley et al. (2016a) concluded that the red algae *Asparagopsis* spp. showed the most potential for $CH_4$ production decrease. Kinley et al. (2016b) further demonstrated that forage with the addition of 2% *Asparagopsis taxiformis* could eliminate $CH_4$ production *in vitro* without negative effects on forage digestibility. In recent animal experiments, reduction of enteric $CH_4$ production by more than 98% was achieved with only 0.2% addition of freeze-dried and milled *Asparagopsis taxiformis* to the to the organic matter (OM) content of feedlot cattle feed (Kinley et al., 2020).

Halogenated, biologically active secondary metabolites are pivotal in the reduction of $CH_4$ induced by Asparagopsis (Abbott et al., 2020). Most of the reduction is ascribed to bromoform ($CHBr_3$) inhibition of the $CH_4$ biosynthetic pathway within methanogens (Machado et al., 2016). $CHBr_3$ as





a natural halogenated volatile organic compound originates from chemical and biological sources including marine phytoplankton and macroalgae (Carpenter et al., 2000; Quack and Wallace, 2003). When emitted to the atmosphere, $CHBr_3$ has an atmospheric lifetime shorter than six months and is often referred to as a very short-lived substance (VSLS). The halogenated VSLSs have drawn considerable interest because of their potential to deplete stratospheric ozone (Engel and Rigby, 2018). Bromoform is the dominant compound among bromine-containing VSLSs emissions, resulting mostly from natural sources (Quack and Wallace, 2003) and to a lesser degree from anthropogenic production (Maas et al., 2019; 2021). With an atmospheric lifetime of about 17 days (Carpenter and Reimann, 2014), $CHBr_3$ can deliver bromine to the stratosphere under appropriate conditions of emission strength and vertical transport (e.g., Aschmann et al., 2009; Liang et al., 2010; Tegtmeier et al., 2015, 2020) and thus contribute to ozone depletion at middle and high altitudes (e.g., Yang et al., 2014; Sinnhuber and Meul, 2015). Global research on enabling large-scale seaweed *Asparagopsis* farming is increasing (Black et al., 2021) as it appears to be one of the most promising options as an antimethanogenic feed ingredient to achieve carbon neutrality in the livestock sector within the next decade (Kinley et al., 2020; Roque et al., 2021). In consequence, the environmental impact of $CHBr_3$ due to *Asparagopsis* farming also needs to be explored and elucidated.

The hypothesis was that large scale cultivation of *Asparagopsis* would not contribute significantly to depletion of the ozone layer. The aim of this study was elucidation of anthropogenic and natural processes that may contribute to $CHBr_3$ emissions inherent in large scale production of *Asparagopisis spp.* and the subsequent impact of $CHBr_3$ release to the atmosphere by using cultivation in Australia as the model. Specific objectives were to inform the industry on: (i) the potential impact of $CHBr_3$ associated with mass production of *Asparagopsis* on atmospheric halogen budgets and ozone depletion; (ii) potential impacts relative to variability in regional climate, atmospheric conditions, and convection trends with different potentials for transport of $CHBr_3$ to stratospheric ozone; (iii) the combined $CHBr_3$ emissions potential of ocean and terrestrial based cultivation of *Asparagopsis* to supply sufficient biomass for up to 50% of beef feedlot and dairy cattle in Australia; and (iv) extrapolation of the impacts of production to requirements on a global scale.

**2. Data and Method**



The potential impact of $CHBr_3$ on the atmospheric bromine budget and stratospheric ozone depletion, associated with *Asparagopsis spp.* mass production was assessed for assumed annual yields and particular production scenarios of macroalgae in Australia. Terrestrial systems cultivation and open ocean cultivation under different harvest conditions, variations of seaweed yield and growth rates for various scenarios and locations were tested as described in the following subsections.

### 2.1 Cultivation Scenarios

The cultivation scenarios in this study assume that sufficient seaweed is grown to supply *Asparagopsis spp.* to 50% of the Australian herds of beef cattle in feedlots (100%: ~$1.0 \times 10^6$) and dairy cows (100%: ~$1.5 \times 10^6$). For a effective reduction of $CH_4$ production from ruminants, a 0.38% addition of freeze-dried and milled *Asparagopsis taxiformis* to the daily feed dry matter intake (DMI) is required (Kinley et al., 2020). This results in daily feed additions of 38 g dry weight (DW) *Asparagopsis* per head of feedlot cattle and 94 g DW *Asparagopsis* per head of dairy cows. In total, the required annual yield amounts to $3.4674 \times 10^4$ Mg DW *Asparagopsis* to supplement the feed of roughly 50% of the Australian feedlot cattle and Australian dairy cows. Assuming that fresh weight (FW) has a DW content of 15%, a total of $2.3116 \times 10^5$ Mg FW *Asparagopsis* needs to be harvested every year.

For a global scenario, we make the functional assumptions that: (i) there would be adoption of 30% of the global feed base to be supplemented with *Asparagopsis* farmed in Australia to reduce ruminant $CH_4$ production worldwide; (ii) *Asparagopsis* would be adopted by 50% of Australia's feedlot and dairy industries; and (iii) this is approximately equivalent to 1% of the global feedlot and dairy herds for the purpose of both assumed magnitude of production and adoption relevant for calculations of supply and emissions. This export scenario requires for 30 times increased production compared to the Australian scenario if all the required *Aspargopsis* was to be cultivated in Australia and an annual harvest of ~1 Tg DW *Asparagopsis* would be needed from Australian waters.

For the future farm distributions in Australia, we assume that *Asparagopsis* will be cultivated in open ocean systems and terrestrial confinement systems (that may include, but not limited to, tanks, raceways, and ponds) located near Geraldton, Triabunna, and Yamba (Figure 1). We assume that



one third of the required annual yield (=$1.1558 \times 10^4$ Mg DW) is grown near Triabunna (T), with
60% in terrestrial systems and 40% in open ocean farms, one third is grown in terrestrial systems
at Yamba (Y), and the last third is grown in the open ocean in Geraldton (G). For comparison of
the environmental impact, we also adopt a tropical scenario where all farms with their total annual
yield of $3.4674 \times 10^4$ Mg DW are assumed to be situated near Darwin.
The emissions of $CHBr_3$ from the macroalgae farms can be derived based on estimates of the
standing stock biomass. For any given farming scenario, the standing stock biomass $B_f$ (g DW) is
a function of time $t$ and can be calculated from the initial biomass $B_i$ (g DW) and the specific
growth rate $GR$ (%/day) according to Hung et al. (2009):

$$B_f(t) = B_i \cdot (1 + GR/100)^t \qquad (1)$$

Terrestrial systems and open ocean cultivation scenarios are assuming a fixed targeted annual yield.
For a given initial biomass and growth rate, the length and frequency of the growth periods per
year need to be chosen accordingly, to achieve the required final yield. Yong et al., (2013) checked
the reliability of different equations for seaweed growth rate determination by comparing the daily
seaweed weight cultivated under optimized growth condition, and the most reliable relationship
between initial and final weight leads to the form of Eq (1). We also applied several growth rates
from 1 to 10% to show the possible influence of this parameter on the overall emissions of the
algae. Average growth rates of *Asparagopsis* ranged from 7 to 13 %/day in samples from tropical
and sub-tropical Australia during short-term experiments (Mata et al., 2017). We used a lower
growth rate of 5% for our scenario to provide an upper estimate of potential $CHBr_3$ emissions.
Note that emissions decrease by 27% when using a growth rate of 7% as demonstrated in section

156  3.1.

Figure 2 provides an example of the variations of standing stock of *Asparagopsis* for the farms of
Geraldton (all open ocean) and Yamba (all terrestrial systems) with a growth rate of 5% per day.
For the open ocean cultures, we assume a scenario of six harvests per year and 60 day growth
periods to obtain the annual yield (Elsom, 2020). For a sensitivity study, we assume an alternative
scenario based on the same initial biomass, but only one harvest per year. As evident from Figure
2, the same annual yield can be achieved with one harvest per year if applying an extended growth
period of 96 days. For the tank cultures, a harvest every 5 days (73 harvests per year) is assumed
as a realistic scenario (Elsom, 2020).

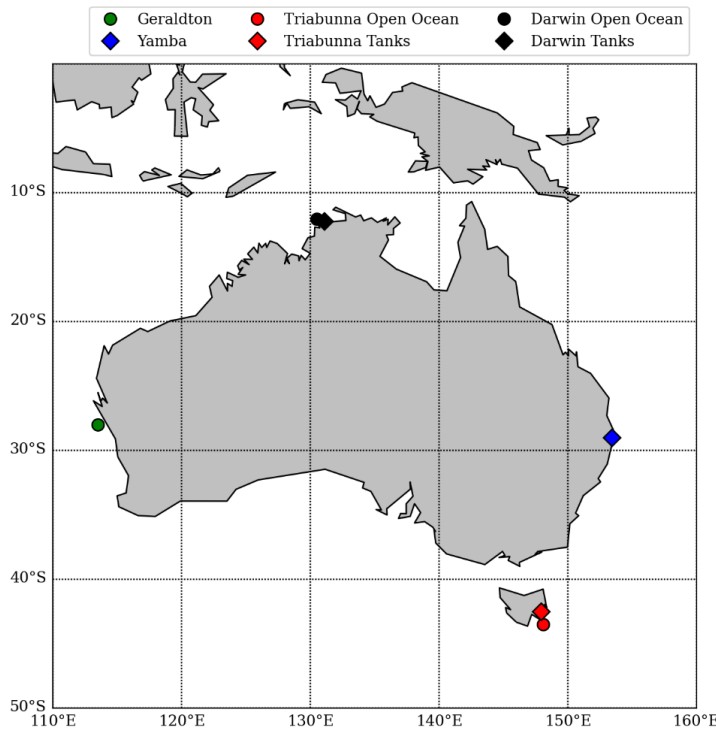

**Figure 1.** Locations of actual and theoretical *Asparagopsis* farms in Geraldton, Triabunna, Yamba,
and Darwin.





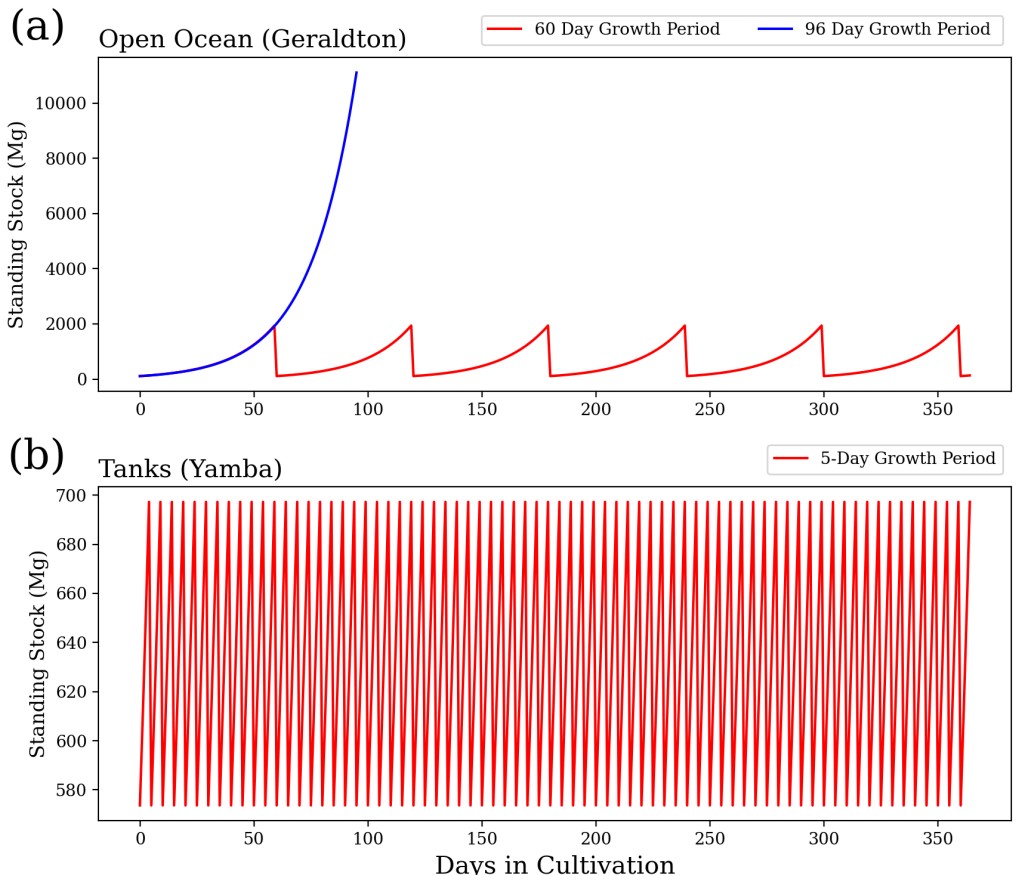

**Figure 2.** Standing stock biomass of *Asparagopsis* cultivation a) in the open ocean for a 60-day growth period and 96 day growth period and b) in terrestrial systems culture for a 5 day growth period. Each of the three scenarios will achieve an annual yield of $1.1558 \times 10^4$ Mg DW.

### 2.2 *Asparagopsis* CHBr₃ release rates

Rates of the $CHBr_3$ content in *Asparagopsis* given in the literature range between 3.4 to 43 mg $CHBr_3$/g DW, with values around 10 mg $CHBr_3$/g DW appearing to be realistic in current cultivation (Burreson and Moore, 1976; Mata et al., 2012, 2017; Paul et al., 2006; Vucko et al., 2017). We assume that *Asparagopsis* strain selection cultivated for feed supplements will lead to high yielding $CHBr_3$ varieties thus we assume augmented $CHBr_3$ production with a mean content





of 21.7 mg CHBr$_3$/g DW (Magnusson et al., 2019) for this study.
Very few values on the CHBr$_3$ release from *Asparagopsis* have been reported in the literature. A
constant release of 1100 ng CHBr$_3$/g DW hr$^{-1}$ was measured for *Asparagopsis armata*
*tetrasporophyte*, which has a CHBr$_3$ content of 14.5 mg CHBr$_3$/g DW (Paul et al., 2006). We
assume a linear scaling between the CHBr$_3$ release rates and the content. Thus, a cultivated
*Asparagopsis* for which we assume 21.7 mg CHBr$_3$/g DW should release around 1646 ng CHBr$_3$/g
DW hr$^{-1}$, a rate which has been confirmed by Marshall et al. (1999). Therefore, for our calculations,
we assume a constant release of 1600 ng CHBr$_3$/g DW hr$^{-1}$ for farmed *Asparagopsis* with a CHBr$_3$
content of 21.7 mg CHBr$_3$/g DW. These content and release rates are higher than for wild stock
algae (Leedham et al., 2013; Nightingale et al., 1995) as the farming aims at high yielding CHBr$_3$
varieties. As available information on this topic is very sparse no variations of the release rate with
life-cycle stages, season, location, or other environmental parameters was used in this study. Also,
the two species *Asparagopsis armata* and *Asparagopsis taxiformis* were treated the same way as
*Asparagopsis spp*., as variations in CHBr$_3$ content and release within or between species are
currently unknown (Mata et al., 2017) and more research on this topic is needed.

196        **2.3 Parameterization of CHBr$_3$ Emission**


The emissions of CHBr$_3$ from farmed macroalgae are a function of the standing stock biomass (in
g DW) and can be calculated with the constant release rate ($R_{CHBr_3}$) of 1600 ng CHBr$_3$/g DW hr$^{-1}$
multiplied with the standing stock. The total release of CHBr$_3$ ($E_{CHBr_3}$) over the complete growth
period of T days is given by the integral over the daily emissions from day 1 to day *T*:
$$E_{CHBr_3} = \int_0^T 24 \cdot B_i \cdot (1 + GR)^t \cdot R_{CHBr_3} dt = 24 \cdot B_i \cdot R_{CHBr_3} \cdot \frac{[(1+GR)^T - 1]}{ln\,(1+GR)} \qquad (2)$$
For our atmospheric impact studies we assume, that all CHBr$_3$ released from the algae is emitted
into the atmosphere at its location of production. An increasing seawater concentration of CHBr$_3$
shifts the equilibrium conditions between seawater and air towards the atmosphere, as CHBr$_3$
easily volatilizes to the atmosphere. Consequently, air-sea exchange acts as a relatively fast loss
process for CHBr$_3$ in surface water. Oceanic sinks can also impact CHBr$_3$, but act on relatively
long timescales. Degradation through halide substitution and hydrolysis results in the ocean sink
CHBr$_3$ half-life of 4.37 years (Hense and Quack, 2009). Thus, most of the CHBr$_3$ contained in





surface seawater is instantly outgassed into the atmosphere without oceanic loss processes playing
a role as confirmed by the modelling study of Maas et al. (2020).
The air-sea exchange of CHBr$_3$ is expressed as the product of its transfer coefficient ($k_w$) and the
concentration gradient ($\Delta c$) (Eq. (3)). The gradient is between the water concentration ($c_w$) and
theoretical equilibrium water concentration ($c_{atm}/H$), where $c_{atm}$ is the atmospheric concentration
and $H$ is Henry's law constant (Moore et al., 1995a; Moore et al., 1995b).
$$F = k_w \cdot \Delta c = k_w \cdot (c_w - \frac{c_{atm}}{H}) \qquad (3)$$
The compound-specific transfer coefficient ($k$w) is determined using the air-sea gas exchange
parameterization of Nightingale et al. (2000) (Eq. (4))
$$k_w = k \cdot \sqrt{Sc}/660 \qquad (4)$$
The transfer coefficient $k$ is a function of the wind speed at 10 m height ($u_{10}$): $k = 0.2u_{10}^2 + 0.3u_{10}$,
and the Schmidt number ($Sc$) is a function of sea surface temperature (SST) from Quack and
Wallace (2003), which is expressed as $Sc = 4662.8 - 319.45 \cdot SST + 9.9012 \cdot SST^2 +$
$0.1159 \cdot SST^3$.
In this study, we use the CHBr$_3$ sea-to-air flux climatology from Ziska et al. (2013) as marine
background emissions. The global emission scenario from Ziska et al. (2013) is a bottom-up
estimate of the oceanic CHBr$_3$ fluxes, generated from atmospheric and oceanic surface ship-borne
*in situ* measurements between 1979 to 2013. Due to the paucity of data the 35 year mean gridded
data set was filled by inter- and extrapolating the *in situ* measurement data. The oceanic emissions
were calculated with the transfer coefficient parameterization of Nightingale et al. (2000) and 6-
hourly meteorological data, which allow a temporal emission variability related to wind and
temperature.

233       **2.4 FLEXPART**


To quantify the atmospheric impact of CHBr$_3$ emissions from macroalgae farming, the Lagrangian
particle dispersion model FLEXPART (Pisso et al., 2019) is used. FLEXPART has been evaluated
extensively in previous studies (e.g., Stohl et al., 1998; Stohl and Trickl, 1999). The model includes
moist convection and turbulence parameterizations in the atmospheric boundary layer and free
troposphere (Forster et al., 2007; Stohl and Thomson, 1999). The European Centre for Medium-
Range Weather Forecasts (ECMWF) reanalysis product ERA-Interim (Dee et al., 2011) with a



241 horizontal resolution of 1° x 1° and 60 vertical model levels is used for the meteorological input

242 fields, providing air temperature, winds, boundary layer height, specific humidity, as well as

243 convective and large-scale precipitation with a 3-hour temporal resolution.

244 We conduct FLEXPART simulations for different emission scenarios as explained in the following

245 and summarized in Table 1:

246 1.) Australian scenarios: $CHBr_3$ emissions from the *Asparagopsis* farming in Geraldton, Triabunna,

247 and Yamba are calculated for an overall annual yield of $3.4674 \times 10^4$ Mg DW according to Equation

248 2. For the terrestrial systems, 5 day growth periods are assumed resulting in 73 harvests per year.

249 For the open ocean, the assumption of different growth periods results in three sub-scenarios a) 6

250 times 60 day growth periods with the first period starting on January 1st (referred to as GTY_O60),

251 b) one 96 day growth period starting on January 1st (GTY_O96_Jan), and c) and another starting

252 on July 1st (GTY_O96_Jul).

253 For the last Australian scenario, we assume that all farms are located around Darwin in the

254 Northern Territory tropics with 6 times 60 day growth periods in the open ocean and 73 times 5

255 day growth periods in the terrestrial systems (Darwin_O60). While this is an unlikely scenario

256 according to current plans, it is useful to demonstrate the influence of potential farming locations

257 on their environmental impact.

258 2.) Global scenarios: Emissions from *Asparagopsis* farming in Geraldton, Triabunna, and Yamba

259 are estimated according to the annual yield, upscaled by a factor of 30 to global requirements.

260 amounting to $1.04 \times 10^6$ Mg DW. Growth periods and harvesting frequencies are set up in the same

261 way as for the Australian scenarios. Short names of the global scenarios are the same as for the

262 Australian scenarios with the additional label 30x.

263 3) Background scenario: Emission from Ziska et al. (2013) for the entire coastal region around

264 Australia defined as all 1°x1° grid cells directly neighbouring the coastline (Ziska_Coast).

265 4.) Extreme event scenarios:  We assume extreme conditions where a hypothetical tropical cyclone

266 causes implausible release of all $CHBr_3$ from the macroalgae farm and water into the atmosphere.

267 We focus on the case study of Geraldton and the tropical cyclone Joyce, which occurred from 6-

268 13 January 2018 around western Australia. We base the amount of available macroalgae biomass

269 on the Australian scenario and assume that the entire $CHBr_3$ content of all *Asparagopsis* at this

270 location is released at once. The two scenarios defined here assume that the tropical cyclone occurs

271 at the end of the 60 day growth period (Geraldton_Ex60) resulting in the release of 41.8 Mg $CHBr_3$





(21.7 mg $CHBr_3$/g DW * 1926 Mg DW) or at the end of the 96 day growth period (Geraldton_Ex96)
resulting in the release of 250.8 Mg CHBr3 (21.7 mg $CHBr_3$ /g DW* $1.1558\times10^4$ Mg DW).

The daily model output is recorded for all simulations. For the extreme event, which assumes the
destruction of a farm (Geraldton-Ex), the 3 hourly output is recorded. For all simulations, except
the background scenario and extreme scenario, trajectories are released from four regions of the
size of: a) Geraldton (open ocean, 11558 ha): 0.1°x0.1°; b) Triabunna (open ocean, 4623 ha):
0.06°x0.06°; c) Triabunna (terrestrial systems, 126 ha): 0.01°x0.01°; and d) Yamba (terrestrial
systems, 210 ha): 0.01°x0.01°. For the tropical and extreme scenarios, trajectories are released
from the Darwin and Geraldton farms, respectively. For the background scenario Ziska_Coast,
trajectories are released from the 1.0°x1.0° grid along the Australian coastline. The amount of
released $CHBr_3$ is evenly distributed among the trajectories and is depleted during the Lagrangian
simulations according to the atmospheric half-life of 17 days (*e*-folding lifetime of 24 days)
(Hossaini et al., 2010; Montzka and Reimann, 2010).







**Table 1.** Detailed information on the scenarios set up for the atmospheric transport simulations
with FLEXPART (Geraldton, Triubanna, and Yamba: GTY)

| Name | | Total Yield (Mg DW) | $CHBr_3$ Emissions (Mg) | Notes | Simulation Period |
|---|---|---|---|---|---|
| **Australian Scenarios** | GTY_O60 | Total: 34674 | Total: 27.3 | 6 harvests (every 60 days) in the open ocean; 73 harvests (every 5 days) in the terrestrial systems. | |
| | GTY_O96_Jan | Open Ocean: G: 11558 T: 4623 Y: - | Open Ocean: G: 9.10 T: 3.64 | 1 harvest (after 96 days) in open ocean; 73 harvests (every 5 days) in terrestrial systems. Growth in open ocean starts from 01.01.2018. | |
| | GTY_O96_Jul | Terrestrial systems: G: - T: 6935 Y: 11558 | Terrestrial systems: T: 5.46 Y: 9.10 | Same as GTY_O96_Jan but with growth in open ocean starting from 01.07.2018. | |
| | Darwin_O60 | Total: 34674 Open Ocean: Darwin: 16181 Terrestrial systems: Darwin:18493 | Total: 27.3 Open Ocean: Darwin: 12.7 Terrestrial systems: Darwin: 14.6 | Same as GTY_O60 but with farms near Darwin | 01.01.2018 - 31.12.2018 2-month spin-up |
| **Global Scenarios** | GTY_O60_30x | Total: $1.04022 \times 10^6$ | Total: 819 | Same as GTY_O60 but with initial biomass and areas 30 times larger. | |
| | GTY_O96_Jan_30x | Open Ocean G: $3.4674 \times 10^5$ T: $1.3869 \times 10^5$ Y: - | Open Ocean G: 273 T: 109.2 | Same as GTY_O96_Jan but with initial biomass and areas 30 times larger. | |
| | GTY_O96_Jul_30x | Terrestrial systems: G: - T: $2.0805 \times 10^5$ Y: $3.4674 \times 10^5$ | Terrestrial systems: T: 163.8 Y: 273 | Same as GTY_O96_Jul but with initial biomass and areas 30 times larger. | |
| | Darwin_O60_30x | Total: $1.04022 \times 10^6$ Open Ocean Darwin: $4.8543 \times 10^5$ | Total: 819 Open Ocean Darwin: 381 Terrestrial systems: Darwin: 438 | Same as Darwin_O60 but with initial biomass and areas 30 times larger. | |





| | | | | | |
|---|---|---|---|---|---|
| | | Terrestrial systems: Darwin: $5.5479\times10^5$ | | | |
| **Background Scenario** | Ziska_Coast | - | 3109 | $CHBr_3$ emission of the coastal region of Australia from Ziska et al. (2013) | |
| **Extreme Scenarios** | Geraldton_Ex60 | Open Ocean: G: 1926 | Open Ocean: G: 41.8 | Extreme event: $CHBr_3$ in Geraldton surface water before harvest is released due to tropical cyclone Joyce (07.01.2018 – 15.01.2018). Harvest period: 60 days. | 9.01.2018 – 9.02.2018 No spin-up |
| | Geraldton_Ex96 | Open Ocean: G: 11558 | Open Ocean: G: 250.8 | Same as Geraldton _Ex60 but with harvest period of 96 days | |

**2.5 Ozone Depletion Potential (ODP)**

The ozone depletion potential (ODP) is defined as the time-integrated potential destructive effect
of a substance to the ozone layer relative to that of the reference substance CFC-11 (Wuebbles,
1983). The ODP is a well-established and extensively used concept traditionally defined for
anthropogenic long-lived halogens. However, the concept has been also applied to VSLSs
(Brioude et al., 2010; Pisso et al., 2010): unlike the ODP for long-lived halocarbons, which is one
constant number, the ODP of a VSLS is a function of time and location of the emissions. This
variable number still describes the time-integrated ozone depletion resulting from a $CHBr_3$ unit
mass emission relative to the ozone depletion resulting from a unit mass emission of CFC-11.
However, the trajectory-derived ODP of e.g. $CHBr_3$ is calculated as a function of location and time
of the potential emissions. As for the classical ODP, and independently of the total amount of
CHBr3 emitted, the time and space dependent ODP describes only its potential but not the actual
damaging effect to the ozone layer. The fraction of originally emitted VSLSs reaching the
stratosphere depends strongly on the meteorological conditions. In particular, it shows a
pronounced seasonality. Here we apply ODP values adapted from Pisso et al. (2010), originally
calculated for a VSLS with a lifetime very similar to that of $CHBr_3$. ODPs for VSLSs are calculated
by means of combining two sources of information: one corresponding to the slow stratospheric
branch and the other to the fast tropospheric branch of transport. The former is uniform for all
species modelled and is based on the calculation of the expected stratospheric residence time of a
Lagrangian particle entering the stratosphere. The latter is based on the probability of stratospheric



injection of a given unit emission of the tracer at the ground. The probability of injection depends
not only on the fraction of air reaching the tropopause but also on the time the air mass takes from
the ground to the tropopause. This is because during the transit of the air mass through the
troposphere, the precursor is chemically degraded, and the solubility of the products leads to mass
loss due to wet deposition.
In this study, we present the ODP-weighted emissions, which combine the information of the ODP
and surface emissions and are calculated by multiplying the $CHBr_3$ emissions with the trajectory-
derived ODP at each grid point. The ODP-weighted emissions provide insight in where and when
$CHBr_3$ is emitted that impacts stratospheric ozone (Tegtmeier et al., 2015). The absolute values
are subject to relatively large uncertainties arising from uncertainties in the parameterization of the
convective transport. Furthermore, the here applied ODP values do not consider product gas
entrainment and provide therefore a lower limit of the impact of $CHBr_3$ on stratospheric ozone.
Taking into account product gas entrainment can lead to 30% higher ODP values (Engel and Rigby,
2018; Tegtmeier et al., 2020), but has no large impact on the here presented comparison of global
ODP-weighted $CHBr_3$ emissions with farm-based ODP-weighted $CHBr_3$ emissions.

**3.  $CHBr_3$ Emission and Atmospheric Mixing Ratio**
**3.1 $CHBr_3$ Emissions**

As shown in Eq. (2), the total $CHBr_3$ emissions are determined by the growth rate, growth period
and initial biomass. For our scenarios based on selected fixed growth rates, the growth periods are
adjusted so that the intended annual yield (~$3.5 \times 10^4$ Mg DW) is achieved. We conduct a sensitivity
study to analyze how much the total emissions change for variations of the length and number of
the growth periods for a fixed annual yield. For this purpose, we compare Geraldton farming for
GTY_O60 (open ocean, six 60 day growth periods) with Geraldton farming for GTY_O96 (open
ocean, one 96 day growth period) and Yamba farming for GTY_O60 (terrestrial systems, 73
growth periods of 5 days). Our estimates reveal that the annual release of $CHBr_3$ from
Asparagopsis is the same for all three case studies (Fig. 3a), indicating that for a fixed annual yield
and growth rate, the culture conditions of open ocean and tank farming are not important for
VSLSs emissions.





A second sensitivity study investigates the variations of $CHBr_3$ emissions for different growth rates
and the same fixed annual yield. For this purpose, we compare Geraldton farming (open ocean,
with an intended annual yield of ~$1.1\times10^4$ Mg DW) for different growth rates varying between 1%
and 10%. The scenario with a 5% growth rate corresponds to Geraldton farming for GTY_O60
(open ocean, six 60 day growth periods), while for the other growth rates the growth periods have
been adjusted to achieve the same annual yield.
The $CHBr_3$ emissions depend strongly on the growth rates (Fig 3b), with emission calculated for
a 1% growth rate being almost 10 times higher than the emissions calculated for a 10% growth
rate. For a lower growth rate, the initial biomass needs to be higher to achieve the targeted seaweed
yield (~$1.1\times10^4$ Mg) after one year and/or the growth period needs to be longer, thus resulting in
larger amounts of biomass in the ocean and higher annual $CHBr_3$ emissions. Vice versa, for higher
growth rates, the annual oceanic biomass is smaller and total emissions are lower.
The overall emissions from the intended Australian seaweed farming of ~$3.5\times10^4$ Mg DW range
from 13.5 Mg (0.05 Mmol) for a 10% growth rate to 134 Mg (0.5 Mmol) per year for a 1% growth
rate. For the growth rates higher than 5%, the differences of $CHBr_3$ emissions are less significant
than those derived for the lower growth rates. In our study, we choose 5% growth rate as
representative, which leads to emissions of ~27 Mg (0.1 Mmol) $CHBr_3$ per year for the targeted
final yield. For the global scenario with an annual yield of ~$1.0\times10^6$ Mg DW (30 times of the
Australian target), the emissions would range from 412 Mg (1.6 Mmol) to 4014 Mg (16 Mmol)
per year, with the annual emission of 810 Mg (3.2 Mmol) for 5% growth rates.
Interestingly, the potential local emissions for all the farming scenarios are generally 3 to 6 orders
of magnitude higher than the background coastal emissions. The maximum climatological
emissions derived from available observations (Ziska_Coast) are around 2000 pmol m$^{-2}$ hr$^{-1}$ for
the coastal waters of Australia, while the emissions from an *Asparagopsis* farm can reach more
than $2.0\times10^6$ pmol (2 μmol) m$^{-2}$ hr$^{-1}$ from a terrestrial system and more than $5.0\times10^5$ pmol m$^{-2}$ hr$^{-1}$
from the open ocean. These differences are to a large degree related to the fact that the
Ziska_Coast is given on a 1.0°x1.0° grid, with high coastal values averaging out over the relatively
wide grid cells, while the values derived for the farms apply to much smaller areas.  Tank emission
rates (0.01°x0.01°) and open ocean farming emission rates (0.1°x0.1°) averaged over a 1°x1° grid
cell result in 200 pmol $CHBr_3$ m$^{-2}$ hr$^{-1}$ and 5000 pmol $CHBr_3$ m$^{-2}$ hr$^{-1}$, respectively, thus being very
similar to the Ziska emissions.

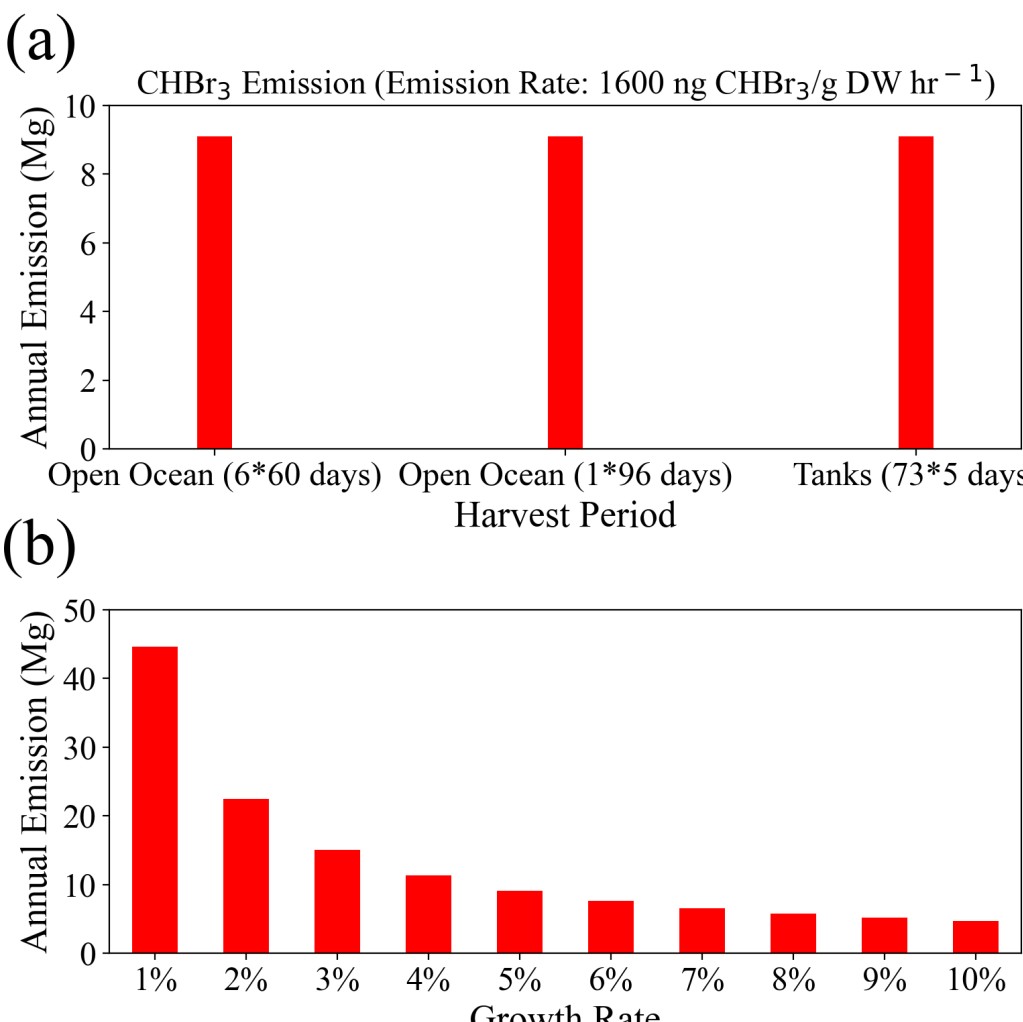

**Figure 3.** The annual release of CHBr$_3$ (Mg yr$^{-1}$) from: a) same growth rate (5%) for different growth periods; and b) under different growth rates, both with a total annual yield of $1.1558 \times 10^4$ Mg DW.


### 3.2 Atmospheric CHBr$_3$ mixing ratio

We use the CHBr$_3$ emissions calculated in section 3.1 to simulate the atmospheric CHBr$_3$ mixing
ratios for each *Asparagopsis* farming scenario. Background CHBr$_3$ levels are calculated based on
the Ziska et al. (2013) Australian coastal emissions (Ziska_Coast). The temporal evolution of
CHBr$_3$ mixing ratio with height shows that the CHBr$_3$ resulting from the Australian farming
scenarios are negligible (see Figure S1) compared to the coastal background emissions of Australia
(Ziska_Coast).
However, for the global scenarios (Figure 4), atmospheric CHBr$_3$ is comparable to CHBr$_3$ resulting
from Australian coastal background emissions, especially near the end of the growth period in the
open ocean. For almost all scenarios (except for GTY_O96 _Jul_30x), the emissions generally
reach higher into the atmosphere in the first three months of the year with enhanced values around
15 km, reflecting the stronger convection during austral summer. For open ocean emissions
occurring during late austral winter (GTY_O96_Jul_30x, Figure 4c), high CHBr$_3$ mixing ratios
are found around September, however at a lower altitude range compared to the equivalent
scenario with open ocean emission occurring during late austral summer (GTY_O96_Jan_30x;
Figure 3b).
The spatial distribution of annual mean CHBr$_3$ at 1 km (Figure 5) further confirms the
insignificance of the signals from the Australian farming scenarios compared with the background
CHBr$_3$ values. For the global scenarios, localized regions of high mixing ratios are found near the
locations of the farms due to the stronger emission. For Darwin_O60_30x, the belt of high mixing
ratios is extending northwestward, due to the prevailing easterlies in the tropics. At higher altitudes
(e.g., 5 km and 15 km; Figure S2-S3), localized high CHBr$_3$ is only found near Darwin for the
Darwin_O60 _30x scenario, reflecting that strong tropical convection is needed to transport short
lived gases to such altitudes.
The results above suggest that in the boundary layer, global scenarios and extreme events could
lead to CHBr$_3$ comparable mixing ratios as those from the background scenario. Only in the global
tropical scenario (Darwin_O60 _30x), CHBr$_3$ mixing ratios, which are larger than the background
values, can be found at high altitudes (Figure 4).
Simulations of the two extreme scenarios (Geraldton_Ex) for 60 and 96 day growth periods are
shown in Figure 6. For the Geraldton_Ex simulations, we assume the implausible scenario that



cyclone Joyce could destroy the farm on the day of harvest in January and the total $CHBr_3$ content
of the *Asparagopsis* stock was simultaneously released to the atmosphere during the event. Both
scenarios lead to significant $CHBr_3$ mixing ratios in the atmosphere, especially at altitudes below
5 km. Among the two scenarios, the Geraldton_Ex96 contributes the larger amount of $CHBr_3$
emission, as the macroalgae experienced a longer growth period, so the biomass was higher and
had accumulated more $CHBr_3$. When averaged over the same period (Jan 9-Jan 26, 2018), the
$CHBr_3$ mixing ratios from Geraldton_Ex96 are much larger than those from Ziska_Coast (Figure
6) below 5 km, and signals with comparable magnitudes are found at 15 km.
As mentioned in section 3.1, the local $CHBr_3$ emissions due to the seaweed cultivation are
generally higher than coastal emission given on 1.0x1.0 grid. However, due to the relatively small
spatial extent of the farms, the emissions quickly dilute in the atmosphere, and the magnitude of
the mixing ratios decline rapidly off the coast and vertically.





**Figure 4.** Altitude-time cross-sections of CHBr₃ averaged over [10°-45° S, 105°-165° E] of CHBr₃ mixing ratio from a) GTY_O60_30x, b) GTY_O96_Jan_30x, c) GTY_O96_Jul_30x, d) Darwin_O60 _30x, and e) Ziska_Coast.

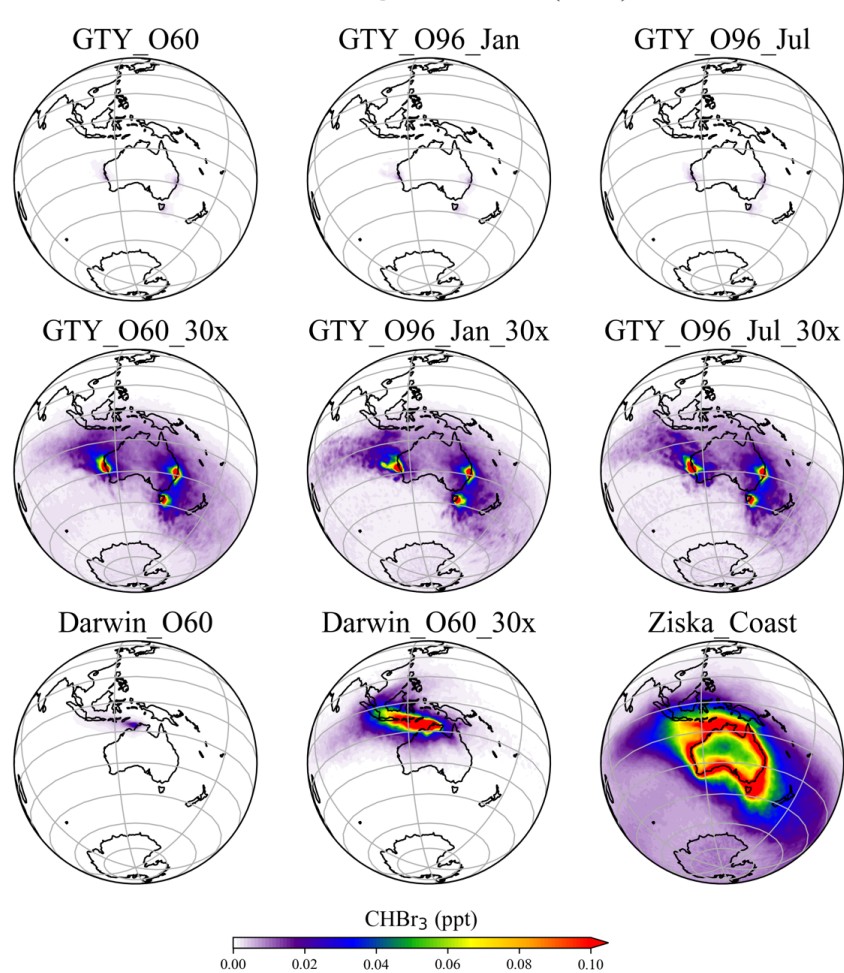

**Figure 5.** Annual mean CHBr₃ spatial distribution from GTY_O60, GTY_O60_30x, GTY_O96_Jan, GTY_O96_Jan_30x, GTY_O96_Jul, GTY_O96_Jul_30x, Darwin_O60, Darwin_O60_30x, and Ziska_Coast at 1 km altitude.



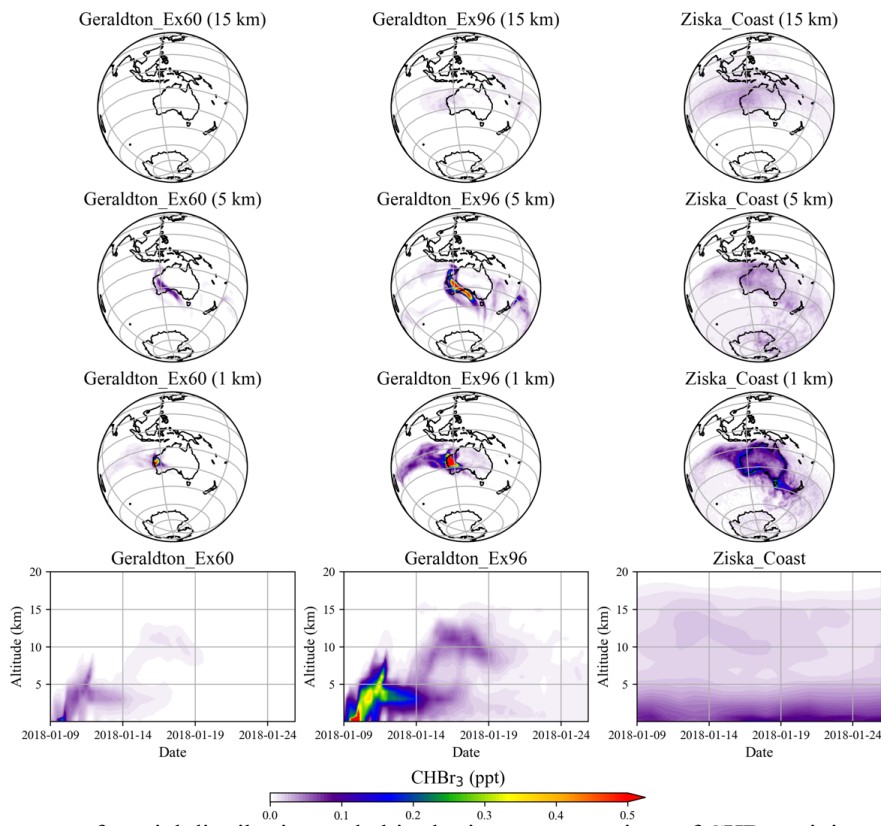

**Figure 6.** 17-day average of spatial distribution and altitude-time cross-sections of $CHBr_3$ mixing
ratio averaged over [10°-45° S, 105°-165° E] for Geraldton_Ex60, Geraldton_Ex96, and
Ziska_Coast.
**4. Ozone depletion potential for $CHBr_3$**
The ODP distribution for the region around Australia, South-East Asia, and the Indian Ocean for
the Southern Hemisphere (SH) summer and winter is shown in Figure 7. The ODP distribution
changes strongly with season as the transport of short-lived halogenated substances such as $CHBr_3$
depends on the seasonal variations of the location of the Intertropical Convergence Zone (ITCZ).
Highest ODP values of 0.5, which imply that any amount (per mass) of $CHBr_3$ released from the
specific location will destroy half as much stratospheric ozone as the same amount of CFC-11
released from this location, are found during July over the Maritime continent and during January
over the West Pacific south of the equator. The northern Australian coastline shows highest ODP


values during January when the thermal equator and the ITCZ are shifted southwards and ODP
values for Yamba and Darwin are 0.26 and 0.29, respectively. The other two locations as well as
all four locations during SH winter, show ODP values of only up to 0.1.
As demonstrated in section 3, the total annual $CHBr_3$ emissions from any location are independent
of the details of the farming practice, however, the ODP-weighted emissions change for the
different scenarios as the growth periods fall into different seasons with varying ODP values. In
general, the scenario of one harvest period in SH summer leads to larger ODP-weighted emissions
when compared to the same biomass harvested throughout the year. In addition to the harvesting
practice, the locations of the farms have a large impact on the efficiency of the $CHBr_3$ transport to
the stratosphere and thus on the ODP-weighted emissions.
The ODP-weighted emissions of $CHBr_3$ for different emission scenarios are shown in Figure 8.
*Asparagopsis* farming at GTY (GTY_O60) leads to additional $CHBr_3$ emissions of up to 2.53 Mg
per year. If all farming (~$3.5 \times 10^4$ Mg DW *Asparagopsis*) occurs in Darwin (Darwin_O60), ODP-
weighted emissions would increase to 6.48 Mg $CHBr_3$ per year. In comparison, all naturally
occurring emissions around the Australian coastline (Ziska_coast) lead to OPD-weighted $CHBr_3$
emissions of 221.52 Mg per year. In consequence, *Asparagopsis* farming in the three locations
Geraldton, Triabunna and Yamba would lead to an increase of the ODP-weighted emissions from
Australian coastal emissions of 1.14%. If all farming would take place in Darwin, ODP-weighted
$CHBr_3$ emissions would increase by 2.93%.
As the global ODP-weighted emissions were estimated to be around $4.0 \times 10^4$ Mg per year
(Tegtmeier et al., 2015), the additional contribution due to the Australian farming scenarios in
GTY or Darwin would be negligible increasing the contribution of $CHBr_3$ emissions to ozone
depletion by 0.006% and 0.016%, respectively. Even if the farming would be upscaled to cover
the global needs (~$1.0 \times 10^6$ Mg DW), the ODP-weighted $CHBr_3$ emissions would only increase to
75 Mg and 195 Mg for farming in GTY (GTY_O60_30x) and Darwin (Darwin_O60_30x),
respectively. Thus produced $CHBr_3$ would increase the current contribution of $CHBr_3$ to
stratospheric ozone depletion by 0.19% and 0.48%, which is again a very small contribution.
To assess the increase of the ODP-weighted $CHBr_3$ emissions under the most extreme and
implausible conditions, we envision the total harvest of one year, which contains 752 Mg (21.7
mg $CHBr_3$/g DW*$3.4674 \times 10^4$ Mg DW) $CHBr_3$, stored in a warehouse of 50 x 25 x 5 m in either
of the four locations.  We assume that the facility is destroyed, and all 750 Mg released to the





atmosphere. Then maximum ODP-weighted $CHBr_3$ emissions would occur for the release in
Darwin during January and amount to 215.9 Mg almost doubling the ODP-weighted coastal $CHBr_3$
emissions of Australia. If the entire content of ~$1.0\times10^6$ Mg *Asparagopsis* DW (21.7 mg $CHBr_3$/g
DW*$1.04022\times10^6$ Mg DW=$2.2573\times10^4$ Mg $CHBr_3$) would be released in Darwin, the additional
contribution of $CHBr_3$ to global ozone depletion could reach 16%.

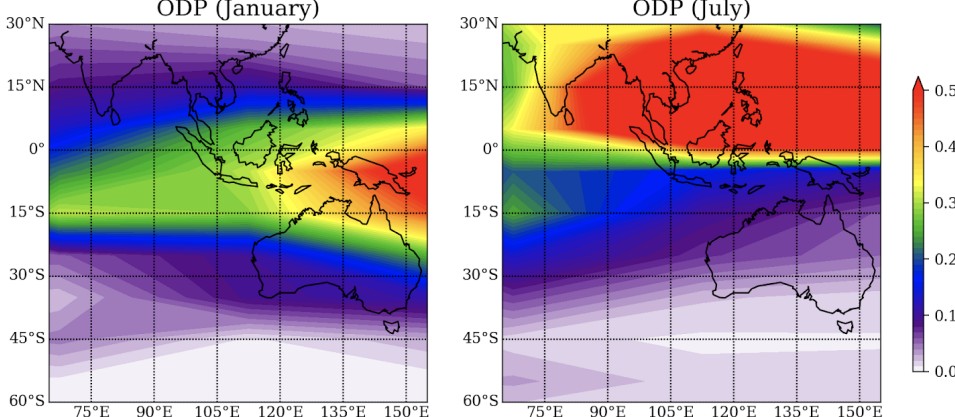

**Figure 7.** Spatial distribution of the ODP in January and July.



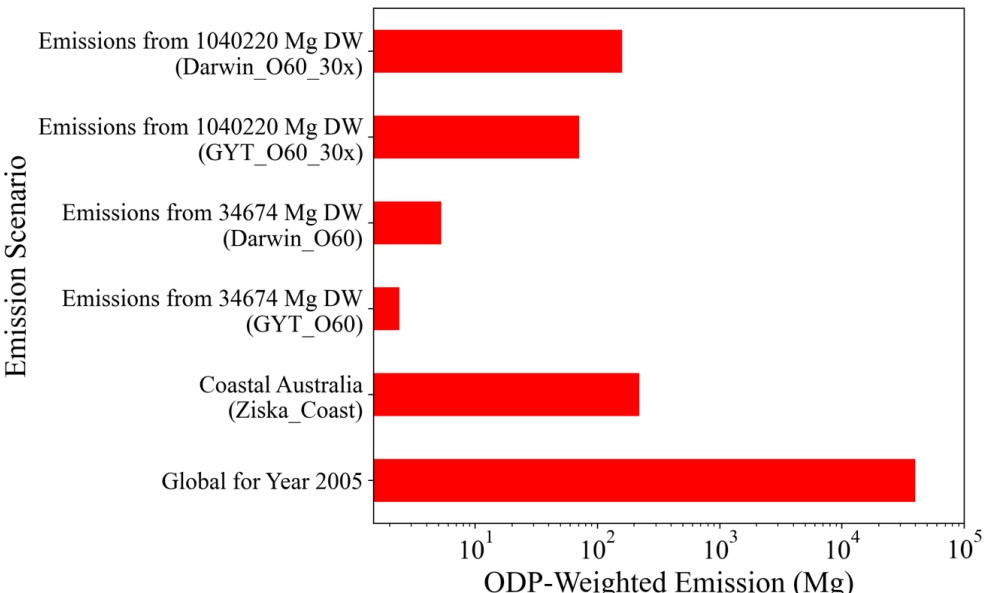

**Figure 8.** The ODP-weighted emissions of CHBr$_3$ for different emission scenarios of *Asparagopsis* farming, incidental content release scenarios and from global and coastal Australian emissions, note that the x-axis is exponential.

## 5. Summary and Conclusions

In this study, we assessed the potential risks of CHBr$_3$ released from *Asparagopsis* farming near Australia for the stratospheric ozone layer by analyzing different cultivation scenarios. We conclude that the intended operation of *Asparagopsis* seaweed cultivation farms with an annual yield of $3.4674 \times 10^4$ Mg DW in either open ocean or terrestrial cultures at the locations Triubanna, Yamba, Geraldton, and Darwin will not impact the ozone layer under normal operating conditions. For Australia scenarios with an annual yield of ~$3.5 \times 10^4$ Mg DW and algae growth rate of 5% per day, the expected annual CHBr3 emission from the considered *Asparagopsis* farms into the atmosphere (~27 Mg, 0.11 Mmol) is less than 0.9% of the coastal Australian emissions (~3109 Mg, 12.3 Mmol). This contribution is negligible from a global perspective by adding less than 0.01% to the worldwide CHBr3 emissions from natural and anthropogenic sources. The overall emissions from the farms would be even smaller with a faster growth rate for the same annual yield. We have assumed a high CHBr$_3$ production of 21.7 mg/g DW from superior strains and





expected lower $CHBr_3$ production of 14 mg/g DW would likewise reduce emissions to the
atmosphere.
The local $CHBr_3$ emissions from the *Asparagopsis* farms could be larger than emissions from
coastal Australia. However, the overall atmospheric impact of the *Asparagopsis* farms is negligible,
as the CHBr3 dilutes rapidly and degrades in the atmosphere under normal weather conditions.
Mixing ratios of $CHBr_3$ are generally dominated by the coastal Australian emissions. In global
scenarios with annual yield $\sim1.0 \times 10^6$ Mg DW, localized $CHBr_3$ mixing ratios comparable to the
background values can be found in the lower troposphere. In the upper troposphere, on the other
hand, mixing ratios larger than background values only appear in the global tropical scenario
(Darwin_O60 _30x). The release of the complete CHBr3 content from the macroalgae to the
environment on very short timescales (e.g., days) due to extreme weather situations could
contribute significant amounts to the atmosphere, especially during times when the standing stock
biomass is relatively large (Geraldton_Ex96). While such extreme scenarios could lead to much
larger mixing ratios than background values, such mass release events are implausible because
even if a farm was totally destroyed the seaweed stock could not instantaneously release all the
accumulated $CHBr_3$. Such scenarios have been included here to evaluate a catastrophic and likely
impossible worst-case scenario.
The impact of $CHBr_3$ from the proposed seaweed farms on the stratospheric ozone layer is assessed
by weighting the emissions with the ozone depletion potential of $CHBr_3$. In total, Australia
scenarios could lead to additional ODP-weighted CHBr3 emissions of up to 2.53 Mg per year with
farms located in Geraldton, Triubana and Yamba. With all farming performed in Darwin
(Darwin_O60), the emitted $CHBr_3$ could reach the stratosphere on shorter time scales and ODP-
weighted emissions would increase to 6.48 Mg, which is less than 0.016% of the global ODP-
weighted emissions. For global tropical scenario (Darwin_O60_30x), the ODP-weighted
emissions amount to 175 Mg, increasing the global ozone depletion by 0.48%, resulting in a very
small contribution.
We note that all data characterizing the potential systems for the production of *Asparagopsis* are
based on few available literature data, lab scale tests and relatively small-scale field trials. This not
only places limitations on the technological representativeness of a future system and the temporal
validity of the study, but also demonstrates importance for directed studies, especially on the
release of $CHBr_3$ from *Asparagopsis* during cultivation. As this understanding evolves so will the

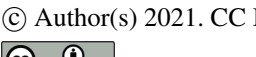



cultivation and processing technologies engineered to conserve the antimethanogenic CHBr$_3$ in
the seaweed biomass which is the primary value feature of *Asparagopsis*. These limitations are
largely mitigated in our study by evaluating various environmental and meteorological conditions
ranging from conservative to most extreme scenarios and by investigating different farming
practices based on various sensitivity studies.

**Data availability**
The CHBr$_3$ emission data and FLEXPART output can be obtained from the authors on request via
BQ (bquack@geomar.de), ST (susann.tegtmeier@usask.ca), or YJ (yue.jia@noaa.gov).

**Author Contributions**
BQ initialized the idea. YJ, BQ, and ST carried out the calculations and analysis. YJ performed
the FLEXPART simulations and produced the figures. YJ, BQ, and ST wrote the manuscript with
the contribution from other co-authors RK and IP.  RK contributed to conceptualization, design,
writing, editing, procurement of funding. All the authors contributed to discussions and revisions
of the manuscript.

**Competing interests**
The authors declare that they have no conflict of interest.
**Acknowledgements**
The authors wish to acknowledge CSIRO, FutureFeed, and Sea Forest for their provision of
technical knowledge, data, and insight into Asparagopsis supply chains in Australia. The authors
would like to thank the European Centre for Medium-Range Weather Forecasts (ECMWF) for the
ERA-Interim reanalysis data and the FLEXPART development team for the Lagrangian particle
dispersion model used in this publication. The FLEXPART simulations were performed on
resources provided by the University of Saskatchewan.





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
