# Peer review of "Potential environmental impact of bromoform from Asparagopsis farming in Australia"

_Atmospheric Chemistry and Physics, 2021_

## Referee Comment (RC1)

Comments on 'Potential environmental impact of bromoform from Asparagopsis farming in Australia' by Jia *et al*. for *Atmos. Chem. Phys.*

Paul Fraser, CSIRO Oceans and Atmosphere, Aspendale, Victoria

Page 1, line 9: delete '4811' – no other co-author post-codes listed

Page 2, line 25: 'which contributes'

Page 2, line 28: 'with its ozone depletion potential'

Page 2, line 29: replace 'halogens' with' halocarbons'

Page 2, line 35: 'significantly impact'

Page 2, line 37: '0.02%' – significant figures implied by 0.016% probably not justified

Page 2, line 37-38: 'The remains are relatively small' does not make sense

Page 2, line 39: 0.5% (as above)

Page 3, line 44:' GHG emissions weighted by radiative forcing'

Page 3, line 47: 'emissions'

Page 3, line 48-49: is this what you mean – 'Total methane ($CH_4$) emissions from ruminant livestock contribute about 18% of total global $CH_4$ emissions'

Page 3, line 51: quote GWP and lifetime data for CH4 from more recent IPCC assessments: IPCC Climate Change 2021

Page 3, line 56: 'inefficient digestion process'

Page 4, line 74: Carpenter and Liss, 2000

Page 4, line 77-78: Engel, Rigby *et al*., 2018

Page 4, line 81: Carpenter, Reimann *et al*., 2014

Page 4, line 83-84:  in the lower and middle stratosphere

Page 4, line 85: Black *et al*. 2021 not in reference list

Page 5, line 115: For an effective

Page 5, line 115: 0.4%? Can two significant Figures (0.38%) be justified; could the experiments distinguish between results for 0.38% and 0.40?

Page 5, line 119: $3.5 \times 10^4$ Mg – 5 significant figures??

Page 5, line 121: $2.3 \times 10^4$ Mg – 5 significant figures??

Page 6, line 135: $1.2 \times 10^4$ Mg

Page 6, line 139: $3.5 \times 10^4$ Mg

Page 6, line 147: Yong et al. (2013)

Page 7, line 166: which farms are actual and theoretical?

Page 8, line 171: $1.2 \times 10^4$ Mg

Page 9, line 180: listed as Magnusson et al. 2020 in reference list

Page 9, line 185: 1650 ng, the Paul et al. release is quoted to 2 significant figures: 1100 ng

Page 9, line 187: OK, 1600 ng, ignore above

Page 9, line 191: were used in this study

Page 10, line 211: Mass et al references are listed as 2019 and 2021

Page 10, line 228: inter-? Interpolation?

Page 11, line 247: $3.5 \times 10^4$ Mg

Page 12, line 285: Montzka, Reimann *et al.* 2010

Page 14, line 298: to that of the reference substance CFC-11 ($CCl_3F$) on a mass emitted basis

Page 14, line 300: long-lived halocarbons

Page 14, line 304: from the same unit mass emission of CFC-11

Page 14, line 307: $CHBr_3$

Page 15 line 326: the ODP values applied here

Page 15, line 328: Engel, Rigby *et al.*

Page 15, line 329-330: impact on the comparison……ODP weighted emissions presented here.

Page 18, line 324: to simulate the enhanced atmospheric $CHBr_3$ mixing ratios (above natural background) for each…

Page 18, line 395: Background $CHBr_3$ levels are calculated…. These background levels derived from Ziska et al. need to be discussed. Do the authors use the latitudinally averaged data (Figure 2 of Ziska at al.). It would be instructive to compare the background $CHBr_3$ levels from Ziska et al. assumed for Triabunna, Tasmania (about 0.5 ppt?) to publicly available (and published) observations at Cape Grim, Tasmania (annual average about 1-2 ppt). The Ziska data compendium are from various laboratories but are not intercalibrated. The Triabunna background $CHBr_3$ level could be as high as 2 ppt – what difference would this make to the flux calculations?

Page 18, line 398: Figure S1 compares calculated $CHBr_3$ mixing ratios due to Asparagopis farming at GTY and Darwin compared to appropriate Ziska mixing ratios (need to state latitude of Ziska data). The Figure shows mixing ratios only, not emissions, so need to state this and then say what this implies about emissions. The background surface $CHBr_3$ mixing

ratios in Figure S1 are 0.01 ppt, this an order of magnitude lower that the Ziska data for Darwin latitudes (minimum 0.5 ppt). Am I missing something here?

Page 18, line 400: Compared to Figure S1, Figure 4 has Ziska coastal surface mixing ratios of 0.05 ppt, 5 times Figure S1??

Page 18, line 409: Figure 5 shows Ziska coastal $CHBr_3$ mixing ratios of 0.1 ppt or higher, at least 2 times Figure 4 and 10Xx Figure S1??

Page 21, Figure 4 caption: Altitude-time cross-sections of $CHBr_3$ mixing ratio averaged over […] from a)…

Page 22, line 456: I would have thought that the ODP distribution also depends on the surface location of the $CHBr_3$ emissions (as well as seasonal transport, location of ITCZ etc). Which emission scenario (strength, location) is used to calculate the ODP distribution?

Page 23, line 479-480: 1.1%, 2.9%

Page 24, line 500: …ODP of $CHBr_3$ in January…. Specify emission scenario?

Page 25, line 514: 3.47

Page 25, line 520: $CHBr_3$

Page 26, line 527, 532, 542: $CHBr_3$

Page 26, line 550: laboratory scale

Page 28, line 608: Battaglia not cited (I may have missed it)

Page 28, line 614: Black et al. 2021 not listed

Page 29, line 663: Herrero et al 2016 not cited (I may have missed it)

Page 31, line 728: Machado et al. 2014 not cited (I may have missed it)

Page 31, line 746: Marshall et al. should be listed before Mata et al.

Page 32, line 775: Montzka, Reimann et al.

Page 33, line 840: Wuebbles at al. 1983 not cited (I may have missed it)

---

## Referee Comment (RC2)

**Review of the manuscript "Potential environmental impact of bromoform from Aspargopsis farming in Australia" by Jia et al., ACPD, 2021.**

The paper presents modeling experiment to evaluate how one of the proposed feeding management alternatives to reduce CH4 emissions from ruminant livestock (i.e., Aspargopsis farming) could impact on the stratospheric ozone layer due to the by-product formation of bromoform (CHBr3). This species is a very short-lived species (VSLS) with a mean lifetime of 17 days in the atmosphere, and consequently, the CHBr3 impact on stratospheric ozone depends on the superposition of source strength and location with the efficiency of convective transport. The paper propose a multiple set of realistic local and global scenarios, as well as the occurrence of some improbable extreme episodes affecting the Australian coast, to evaluate a representative range of the overall ozone depletion potential (ODP) of bromoform emissions from oceanic and terrestrial cultivation approaches, and compare them with the impact of coastal natural bromoform emissions. The work is very well-planed and provides a realistic and clear evaluation of the magnitude of one of the environmental consequences of promoting Aspargopsis production in Australia, and determine that even in the worse possible scenario, the negative impact of the additional farming-released bromoform are very small in comparison with the natural contribution from the ocean. The methodology and results are generally well presented, although some clarification is required as described below. I suggest the paper is accepted for publication after the following issues have been solved:

**Main Comments:**

**1a. Ozone Depleting Potentials (ODPs): Concept and Implications**
Section 2.5 briefly describes the ODP concept and how it has been adapted to evaluate the ODP impact of VSLS due to their variable distribution in the troposphere. However, given the importance of the ODP fields used to determine the bromoform ODP-weighted emissions presented in this work, I found that more details (and results discussion in Section 4) should be given. In particular, the authors based their analysis on the ODP spatiotemporal study performed by (Pisso et al., 2010) using

the same FLEXPART model, but no mention is provided about other approaches to determine the Stratospheric ODP (SODP) for long-lived species that are know to affect both tropospheric and stratospheric ozone (Claxton et al., 2019; Zhang et al., 2020), and why it is important to distinguish the tropospheric and stratospheric ozone impacts of CHBr3. Page 15, Lines 326-329 is the only place in the text where I found explicit mention that the product gas contribution of VSLS degradation is not being considered, which is reasonable as the proposed methodology considers only the exponential decay of the emitted source gases. However, this should be at least highlighted again in the conclusions and if possible, an estimation of the magnitude of the neglected tropospheric impact of VSLS product gases and/or how the modeling ozone changes depend on the treatment of VSLS product gases (i.e., Fernandez et al., 2021) could be given.

**1b. Ozone Depleting Potentials (ODPs): Methodology**

The ODP for bromoform is computed by comparing the ozone destruction of CHBr3 compared with the ozone destruction produced by an equivalent mass of CFC-11. However, no CFC-11 sensitivity is mentioned to have been performed for this study. Thus, it is not clear if Fig. 7 is a direct result of the modeling simulations performed in this work, or it is taken from Pisso et al., (2010). If the later is the case (which I believe it is), then, this should be expressed more clearly in the text and proper reference to this study should be given in the caption of Fig. 7. Page 14, Lines 311-313 explicit says that "ODPs for VSLSs are calculated by means of combining two sources of information: one corresponding to the slow stratospheric branch and the other to the fast tropospheric branch of transport". First, how the tropopause location is determined in the study? Second, is it possible to quantify the contribution of these two branches, and could this be taken as an approximation of the tropospheric and stratospheric influence of CHBr3 farming emissions? Note that one of the main results of the paper is that ODP-weigthed CHBr3 contribution from Aspargopsis farming would be, at most, less than 1% of the natural CHBr3 value (i.e., the Ziska_Coast scenario); thus properly showing how the ODP values were computed for this particular VSLS should be clear.

Minor Comments:

GENERAL: The number of significant digits used when reporting numbers should be revised throughout the hole text.

P2,L36: What do you mean by "the remains are relatively small"?

P2,L37: "less than 0.016%" ... is this significant different to less than 0.02%?

P2,L39: "by 0.48%" ... of its initial value, or up to 0.48%?

P4,L88: "In consequence, the environmental impact of CHBr3 ... *needs to be explored and elucidated*". As detailed in the main comment, the authors should explicit mention that VSLS influence both the troposphere and stratosphere, and that here only the stratospheric impact is considered.

P4,L94: I found the paper very informative not only to industry, but also to policy makers and the scientific community.

P5,L119: $3.4674 \times 10^4$. Does this number have 5 significant digits? Please clarify and make it consistent throughout the text.

P5,L128-130: How did you get the 30 times scaling factor to extrapolate from Australian Aspargopsis production to Global production? And how did you get the 1 Tg DW value? (I could not get that value by multiplying the informed data ... I must have missed something).

Figure 1: The lat,lon region shown in the Figure is smaller than the rectangle used for computing the average of CHBr3 mixing ratio in Figs. 4 and 6.

P10,L233: Considering extending the subsection title so it includes the description of the different scenarios. In addition, by looking at Table 1 it is evident that the study was performed for meteorological conditions of year 2018 ... But I could not find where in the text this is described (I might have missed it).

P11,L263 and Table 1: The total CHBr3 emission within the background scenario considers the well-established Ziska emission inventory, and is mentioned to consider "all 1º×1º grid cells directly neighboring the coastline", which accounts for 3109 Mg (Table 1). How large are the Ziska emissions for a small region of the size of area of Geraldton, Triabunna or Yamba? Similarly, how large are the Ziska_coast emissions if they are compared to the total Ziska emission on the Australian domain [10°-45° S, 105°-165° E] if both coastal and open-ocean grid-cells are considered?

P14,L311: The 20 days lifetime of the VSLS species considered in Pisso et al., (2010), should be mentioned here.

P15,L321-323: "In this study, we present the ODP-weighted emissions, which combine the information of the ODP and surface emissions and are calculated by multiplying the CHBr3 emissions with the trajectory-derived ODP at each grid point". Does Pisso et al., (2010) provide independent ODP values for each model grid-point and individual trajectory? Please see my main comment regarding this point.

P15,L344 and Fig. 3a: The figure is fine, and is clear that the annual emission for the different growth periods are equivalent, but the text seems to imply that this is a new result of the study. However, these equivalent values is just a confirmation of the assumed condition that all farming scenarios for Australia must have the same total emission. This should be clarified in the text.

P16,L363: "which leads to emissions of 27 Mg (0.1 Mmol) CHBr3 per year for the targeted final yield". How do you relate this 27 Mg CHBr3 per year with the aprox. 9 Mg CHBr3 annual emission derived from Fig. 3a? Shouldn't this values be identical? Is it needed to multiply by the bromine atomicity of bromoform (3)?. Please make it clear.

Figure 3 caption: "… under different growth rates *and similar initial biomass and growth period*". Please make the caption as informative as possible.

P19,L430: "and signals with comparable magnitudes are found at 15 km". The magnitudes are comparable, but I expect this signals affect much

smaller regions due to the localized source. Is this the case? If so, please make it explicit for the reader.

Figs. 4 and 5: Is the color scale maximum value correct? i.e. 0.05 ppt for Fig. 4 and 0.10 ppt for Fig. 5? How large are the maximum values within the MBL? I would expect them to be much larger than the maximum value of the scale. The caption of Fig. 4 should also explicitly indicate that it refers to Global scenarios.

Figure 7: If the units of the scale is a relative value between 0 and 1, please make it explicit.

Figure 8: The bottom-most bar presenting values for the Global Emission, for which of the global scenarios apply?

Language editing comments:

GENERAL: A language style revision should be performed to the whole text (as well as figure captions), mainly on the unification of past, present and future terms (is, was, will) into a common verbal tense.

P2,L26-30: Split the sentence.

P2,L30: DW acronym is not used again in the abstract.

P3,L48: Two blank spaces.

P3,L64: rephrase "showed the most potential for CH4 production decrease".

P9,L189: What do you mean by "as the farming aims at high yielding CHBr3 varieties"?

P10,L214: "the gradient is between" … it is computed between? It is computed considering …?

P15, L323-324: "The ODP-weighted emissions provide insight in where and when CHBr3 is emitted that impacts stratospheric ozone (Tegtmeier et al., 2015)". Not sure if the sentence is properly written. Please rephrase.

P15,L329-330: "but has no large impact on the here presented comparison of global ODP-weighted CHBr3 emissions with farm-based ODP-weighted CHBr3 emissions.". Please rephrase.

P19,L423: The authors use the terms "destroy" to refer to the impact of cyclone Joyce on the Australian coast. Please consider using a different wording (here and elsewhere).

P23,L488: remove "again"

P25,L517,520: (here and elsewhere). Use subindex for 3 in CHBr3.

P26,L525: "The local CHBr3 emissions from the Asparagopsis farms could be larger than emissions from coastal Australia." The term "local" here is correct, but seems hidden in the sentence and could be reinforced.

References:
Claxton, T., R. Hossaini, O. Wild, M.P. Chipperfield, and C. Wilson, On the regional and seasonal ozone depletion potential of chlorinated very short-lived substances, Geophys. Res. Lett., 46(10), 5489–5498. doi:10.1029/2018GL081455, 2019.

Fernandez, R.P., J.A. Barrera, A.I. López-Noreña, D.E. Kinnison, J. Nicely, R.J. Salawitch, P.A. Wales, B.M. Toselli, S. Tilmes, J.-F. Lamarque, C.A. Cuevas, and A. Saiz-Lopez, Intercomparison between surrogate, explicit and full treatments of VSL bromine chemistry within the CAM-Chem chemistry-climate model, Geophys. Res. Lett., 48(4), doi:10.1029/2020GL091125, 2021.

Pisso, I., P.H. Haynes and K. S. Law, Emission location dependent ozone depletion potentials for very short-lived halogenated species, Atmos. Chem. Phys., 10, 12025–12036, https://doi.org/10.5194/acp-10-12025-2010, 2010.

Zhang, J., D.J. Wuebbles, D.E. Kinnison, and A. Saiz-Lopez, Revising the ozone depletion potentials metric for short-lived chemicals such as CF3I and CH3I. J. Geophys. Res. Atmos., 125, e2020JD032414. https://doi.org/10.1029/2020JD032414. 2020.

---

## Author Comment (AC1)

Dear Reviewers

We would like to express our sincere gratitude to the reviewers for your effort to improve our manuscript. Based on your comments, we've revised our manuscript accordingly with changed parts marked red.

The following is the point-to-point response with reviewers' comments in bold and the responses italic.

**Comment on acp-2021-800**
**Paul J. Fraser (Referee)**
**Referee comment on "Potential environmental impact of bromoform from Asparagopsis farming in Australia" by Yue Jia et al., Atmos. Chem. Phys. Discuss., https://doi.org/10.5194/acp-2021-800-RC1, 2022**
**Technical comments attached: Jia et al....**
**This is an important paper. CHBr$_3$ is a potent ODS and is produced in substantial quanties in the production of seaweed supplements to the diets of ruminants to suppress their CH4 production. If adopted widely, this technologhy could substantially reduce ruminant CH4 emissions which are a significant component of global CH4 emissions. The paper address the important concept for short-lived ODSs that the impact on the ozone layer is dependent on the location of the emissions. The paper demonstrated the production of the necessary supplements to feed the global ruminant levels does not significantly deplete stratospheric ozone - the technology is 'ozone safe'.**
**I have a technical issue with the assumed/calculated levels of CHBr3 resulting largely from coastal regions and natural seaweeds. I think the Zafra et al. data, which are a compendium of CHBr3 data from several laboratories, and are not intercalibrated (Zaffra et al. recognize this problem and have indicated it will be addressed in future studies) and potentially underestimate background levels of CHBr3 in coastal regions. This seems to be the case in Tasmania (one of the study regions) where measured bachground CHBr$_3$ levels from the AGAGE program (not part of the Zaffra data, but arguable the best measured/calibrated CHBr3 data set available) seem to be up to a factor of 3 higher than the Zaffra et al. data. Is this important? - the authors need to address this.**
*A: As shown in Eq (3) in the manuscript, the flux is calculated as the product of its transfer coefficient ($k_w$) and the concentration gradient ($\Delta c$), which is computed between the water concentration ($c_w$) and theoretical equilibrium water concentration ($c_{atm}/H$), the flux would be even weaker if $c_{atm}$ increases, thus including such higher atmospheric background values in the Ziska methodology would not really increase the Ziska fluxes. On the other hand, if stronger coastal fluxes are applied, the conclusions will still hold.*
*To address this concern, we added several sentences in the discussion section "Another point to note is that Ziska emission potentially underestimate background levels of CHBr$_3$ in coastal regions (Ziska et al., 2013), e.g. CHBr$_3$ measurement in Cape Grim, which is close to Triabunna, shows much larger CHBr3 mixing ratio (~1.5 ppt, Dunse et al., 2020). However, including such*

*higher atmospheric background values in the Ziska methodology would not really increase the fluxes as the flux is driven by air-sea gradient (Eq. 3). Our conclusions will still hold if stronger coastal emission is applied, as it will increase the background CHBr3 mixing ratios.”*

**The authors need to review information on CHBr$_3$ atmospheric lifetime data and ozone impacts in the latest (2021) assessments of climate change (IPCC) and ozone depletion (UNEP)**
*A: The latest ozone assessment is 2018 version (the 2022 version isn't released yet), in which the CHBr$_3$ lifetime doesn't change too much compared to previous ones. We updated the information of CHBr$_3$ lifetime by adding the reference (Engel and Rigby et al., 2018).*

Please also note the supplement to this comment:
https://acp.copernicus.org/preprints/acp-2021-800/acp-2021-800-RC1-supplement.pdf

**Page 1, line 9: delete '4811' – no other co-author post-codes listed**
*A: deleted*

**Page 2, line 25: 'which contributes'**
*A: revised*

**Page 2, line 28: 'with its ozone depletion potential'**
*A: revised*

**Page 2, line 29: replace 'halogens' with' halocarbons'**
*A: replaced*

**Page 2, line 35: 'significantly impact'**
*A: revised*

**Page 2, line 37: '0.02%' – significant figures implied by 0.016% probably not justified**
*A: revised*

**Page 2, line 37-38: 'The remains are relatively small' does not make sense**
*A: The sentence has been revised as "The impact of remaining farming scenarios is also relatively small".*

**Page 2, line 39: 0.5% (as above)**
*A: revised*

**Page 3, line 44:' GHG emissions weighted by radiative forcing'**
*A: revised*

**Page 3, line 47: 'emissions'**
*A: revised*

**Page 3, line 48-49: is this what you mean – 'Total methane (CH4) emissions from ruminant livestock contribute about 18% of total global CH4 emissions'**
*A: Sorry for the confusion, the sentence is revised as "Total GHG emissions (e.g., $CH_4$) from ruminant livestock contribute about 18% of the total global carbon dioxide equivalent ($CO_2$-eq) inventory…"*

**Page 3, line 51: quote GWP and lifetime data for $CH_4$ from more recent IPCC assessments: IPCC Climate Change 2021**
*A: New reference of the sixth IPCC assessment is quoted.*

**Page 3, line 56: 'inefficient digestion process'**
*A: revised*

**Page 4, line 74: Carpenter and Liss, 2000**
**Page 4, line 77-78: Engel, Rigby et al., 2018**
**Page 4, line 81: Carpenter, Reimann et al., 2014**
*A: All the reference issues above are revised.*

**Page 4, line 83-84: in the lower and middle stratosphere**
*A: revised*

**Page 4, line 85: Black et al. 2021 not in reference list**
*A: added*

**Page 5, line 115: For an effective**
*A: revised*

**Page 5, line 115: 0.4%? Can two significant Figures (0.38%) be justified; could the experiments distinguish between results for 0.38% and 0.40?**
**Page 5, line 119: 3.5 x 104 Mg – 5 significant figures??**
**Page 5, line 121: 2.3 x 104 Mg – 5 significant figures??**
**Page 6, line 135: 1.2 x 104 Mg**
**Page 6, line 139: 3.5 x 104 Mg**
*A: The significant figures all through the manuscript were revised. Rounded values with at most two significant figures are used for description parts, except for specific calculations (e.g. Table 1).*

**Page 6, line 147: Yong et al. (2013)**
*A: revised*

**Page 7, line 166: which farms are actual and theoretical?**
*A: the caption of figure 1 is revised as "Locations of actual Asparagopsis farms in Geraldton, Triabunna, Yamba, and theoretical farms in Darwin."*

**Page 8, line 171: 1.2 x $10^4$ Mg**
*A: revised, see above*

**Page 9, line 180: listed as Magnusson et al. 2020 in reference list**
*A: corrected.*

**Page 9, line 185: 1650 ng, the Paul et al. release is quoted to 2 significant figures: 1100 ng**
**Page 9, line 187: OK, 1600 ng, ignore above**

**Page 9, line 191: were used in this study**
*A: revised*

**Page 10, line 211: Mass et al references are listed as 2019 and 2021**
*A: corrected*

**Page 10, line 228: inter-? Interpolation?**
*A: revised as "...filled by interpolating and extrapolating..."*

**Page 11, line 247: 3.5 x 104 Mg**
*A: revised*

**Page 12, line 285: Montzka, Reimann et al. 2010**
*A: revised*

**Page 14, line 298: to that of the reference substance CFC-11 (CCl3F) on a mass emitted basis**
**Page 14, line 300: long-lived halocarbons**
**Page 14, line 304: from the same unit mass emission of CFC-11**
*A: revised*

**Page 14, line 307: $CHBr_3$**
*A: revised*

**Page 15 line 326: the ODP values applied here**

*A: revised*

**Page 15, line 328: Engel, Rigby et al.**

*A: revised*

**Page 15, line 329-330: impact on the comparison……ODP weighted emissions presented here.**

*A: revised*

**Page 18, line 324: to simulate the enhanced atmospheric CHBr3 mixing ratios (above natural background) for each…**

*A: revised*

**Page 18, line 395: Background CHBr3 levels are calculated…. These background levels derived from Ziska et al. need to be discussed. Do the authors use the latitudinally averaged data (Figure 2 of Ziska at al.). It would be instructive to compare the background CHBr3 levels from Ziska et al. assumed for Triabunna, Tasmania (about 0.5 ppt?) to publicly available (and published) observations at Cape Grim, Tasmania (annual average about 1-2 ppt). The Ziska data compendium are from various laboratories but are not intercalibrated. The Triabunna background CHBr3 level could be as high as 2 ppt – what difference would this make to the flux calculations?**

*A: To address this comment, we added a short discussion in Section 5, see the response to the main comment above.*

**Page 18, line 398: Figure S1 compares calculated CHBr3 mixing ratios due to Asparagopsis farming at GTY and Darwin compared to appropriate Ziska mixing ratios (need to state latitude of Ziska data). The Figure shows mixing ratios only, not emissions, so need to state this and then say what this implies about emissions. The background surface CHBr3 mixing ratios in Figure S1 are 0.01 ppt, this an order of magnitude lower that the Ziska data for Darwin latitudes (minimum 0.5 ppt). Am I missing something here?**
**Page 18, line 400: Compared to Figure S1, Figure 4 has Ziska coastal surface mixing ratios of 0.05 ppt, 5 times Figure S1?**
**Page 18, line 409: Figure 5 shows Ziska coastal CHBr3 mixing ratios of 0.1 ppt or higher, at least 2 times Figure 4 and 10Xx Figure S1??**

*A: The differences mentioned in the above 3 comments are from the way how the results are shown (mainly because of averaging). Figure 5 shows the spatial distribution of CHBr₃ mixing ratios, while Figure 4 and Figure 1S show reginal averaged mixing ratios, resulting in smaller mean values. The mixing ratios due to Ziska emission in Figure 4 and Figure 1S (bottom panels) are*

*actually the same, we chose different color scale in the two figures to make the signals of other scenarios more "visible" as the emission magnitudes in Figure 4 are 30 times as in Figure 1S.*

Page 21, Figure 4 caption: Altitude-time cross-sections of CHBr3 mixing ratio averaged over […] from a)…
*A: revised*

Page 22, line 456: I would have thought that the ODP distribution also depends on the surface location of the CHBr$_3$ emissions (as well as seasonal transport, location of ITCZ etc). Which emission scenario (strength, location) is used to calculate the ODP distribution?
*A: The ODP of VSLS is a function of time and location of the emissions, the seasonal transport and location of ITCZ have already implied the location-dependence. The ODP values used in our study are taken from Pisso et al. (2010), in which a map of ODP was created by calculating the ODP for each emission grid globally. The strength of the emission does not matter in calculating the ODP for VSLS as it is the fraction of parcels into the stratosphere released from a certain grid.*

Page 23, line 479-480: 1.1%, 2.9%
*A: revised*

Page 24, line 500: …ODP of CHBr3 in January…. Specify emission scenario?
*A: In Figure 7, the ODP values are from Pisso et al. (2010), the caption has been revised as "Figure 7 Spatial distribution of the ODP in January and July from Pisso et al. (2010), plotted with interval of 0.01". The corresponding description in section 4 is also revised by adding the reference.*

Page 25, line 514: 3.47
*A: revised*

Page 25, line 520: CHBr3
Page 26, line 527, 532, 542: CHBr3
*A: corrected*

Page 26, line 550: laboratory scale
*A: revised*

Page 28, line 608: Battaglia not cited (I may have missed it)
Page 28, line 614: Black et al. 2021 not listed
Page 29, line 663: Herrero et al 2016 not cited (I may have missed it)
Page 31, line 728: Machado et al. 2014 not cited (I may have missed it)
Page 31, line 746: Marshall et al. should be listed before Mata et al.

Page 32, line 775: Montzka, Reimann et al.

Page 33, line 840: Wuebbles at al. 1983 not cited (I may have missed it)

*A: These reference issues have been addressed with missing references added, not cited reference removed, format and order adjusted.*

---

## Author Comment (AC2)

Dear Reviewers

We would like to express our sincere gratitude to the reviewers for your effort to improve our manuscript. Based on your comments, we've revised our manuscript accordingly with changed parts marked red.

The following is the point-to-point response with reviewers' comments in bold and the responses italic.

**Review of the manuscript "Potential environmental impact of**
**bromoform from Aspargopsis farming in Australia" by Jia et al.,**
**ACPD, 2021.**
**Rafael Pedro Fernandez (Referee)**
**Referee comment on "Potential environmental impact of bromoform from Asparagopsis farming in Australia" by Yue Jia et al., Atmos. Chem. Phys. Discuss., https://doi.org/10.5194/acp-2021-800-RC2, 2022**
**The paper presents modeling experiment to evaluate how one of the proposed feeding management alternatives to reduce CH4 emissions from ruminant livestock (i.e., Aspargopsis farming) could impact on the stratospheric ozone layer due to the by-product formation of bromoform (CHBr3). This species is a very short-lived species (VSLS) with a mean lifetime of 17 days in the atmosphere, and consequently, the CHBr3 impact on stratospheric ozone depends on the superposition of source strength and location with the efficiency of convective transport. The paper proposes a multiple set of realistic local and global scenarios, as well as the occurrence of some improbable extreme episodes affecting the Australian coast, to evaluate a representative range of the overall ozone depletion potential (ODP) of bromoform emissions from oceanic and terrestrial cultivation approaches, and compare them with the impact of coastal natural bromoform emissions.**
**The work is very well-planed and provides a realistic and clear evaluation of the magnitude of one of the environmental consequences of promoting Aspargopsis production in Australia, and determine that even in the worse possible scenario, the negative impact of the additional farming-released bromoform are very small in comparison with the natural contribution from the ocean. The methodology and results are generally well presented, although some clarification is required as described below. I suggest the paper is accepted for publication after the issues/comments in the attached file have been solved.**
Please also note the supplement to this comment:
https://acp.copernicus.org/preprints/acp-2021-800/acp-2021-800-RC2-supplement.pdf

**Main Comments:**
**1a. Ozone Depleting Potentials (ODPs): Concept and Implications**
**Section 2.5 briefly describes the ODP concept and how it has been adapted to evaluate the ODP impact of VSLS due to their variable distribution in the troposphere. However, given the importance of the ODP fields used to determine the bromoform ODP-weighted emissions presented in this work, I found that more details (and results discussion in Section 4) should be given. In particular, the authors based their analysis on the ODP spatiotemporal study**

performed by (Pisso et al., 2010) using the same FLEXPART model, but no mention is provided about other approaches to determine the Stratospheric ODP (SODP) for long-lived species that are known to affect both tropospheric and stratospheric ozone (Claxton et al., 2019; Zhang et al., 2020), and why it is important to distinguish the tropospheric and stratospheric ozone impacts of CHBr$_3$.

A: The ODP values calculated in Pisso et al. (2010) are SODP as only the impact on stratospheric ozone was considered. To address this comment, we revised the corresponding part in the introduction by adding "Once released into the atmosphere, degraded halogenated VSLSs can catalytically destroy ozone in the troposphere and stratosphere, thus drawing them considerable interest (Engel and Rigby et al., 2018; Zhang et al., 2020).".

 Also, in Sec 2.5, the following sentences are added "The ODP for VSLSs can be derived from chemistry-climate or chemistry transport models simulating the changes of ozone due to certain compound (Claxton et al., 2019; Zhang et al., 2020). The trajectory-derived ODP of VSLSs such as CHBr$_3$ is calculated as a function of location and time of the potential emissions (Brioude et al., 2010; Pisso et al., 2010). As for the traditional ODP concept, the time and space dependent ODP describes only the potential of a compound but not its actual damaging effect to the ozone layer and is independent of the total emissions. It is noteworthy that many VSLSs including CHBr$_3$ can impact ozone in the troposphere and stratosphere. As ODPs are used to assess stratospheric ozone depletion only, the contribution of VSLSs to tropospheric ozone destruction needs to be excluded when calculating their ODP (Pisso et al., 2010; Zhang et al., 2020). The trajectory based ODP from Pisso et al. (2010) used in this study, considers only the impact of CHBr$_3$ on the stratospheric ozone instead of the ozone column."

**Page 15, Lines 326-329 is the only place in the text where I found explicit mention that the product gas contribution of VSLS degradation is not being considered, which is reasonable as the proposed methodology considers only the exponential decay of the emitted source gases. However, this should be at least highlighted again in the conclusions and if possible, an estimation of the magnitude of the neglected tropospheric impact of VSLS product gases and/or how the modeling ozone changes depend on the treatment of VSLS product gases (i.e., Fernandez et al., 2021) could be given.**

A: In Section 5, we added the following discussion to highlight the missing of product gas contribution "The ODP used in this study, does not include the impact of VSLS product gases. Previous modelling studies have highlighted the role of product gas treatment and their impact on the stratospheric halogen budget (e.g., Fernandez et al., 2021). Including product gas entrainment can lead to up to 30% larger ODP values for CHBr$_3$ (Engel and Rigby et al., 2018; Tegtmeier et al., 2020), thus the ODP-weighted emissions presented here can be up to 30% larger. However, this does not affect our assessment of the potential importance of cultivation induced CHBr$_3$ as the ratios of the impact of each scenario compared with the global ODP-weighted emission remain the same."

**1b. Ozone Depleting Potentials (ODPs): Methodology**
**The ODP for bromoform is computed by comparing the ozone destruction of CHBr3 compared with the ozone destruction produced by an equivalent mass of CFC-11. However, no CFC-11 sensitivity is mentioned to have been performed for this study. Thus, it is not clear if Fig. 7 is a direct result of the modeling simulations performed in this work, or it is taken from Pisso et al., (2010). If the later is the case (which I believe it is), then, this should**

**be expressed more clearly in the text and proper reference to this study should be given in the caption of Fig. 7.**

*A: In this study, the ODP is taken from Pisso et al., (2010). To clarify this, caption of Fig 7 is revised as "Spatial distribution of the ODP in January and July from Pisso et al. (2010), plotted with interval of 0.01". And the beginning of Sec 4 is also revised as "The ODP distribution from Pisso et al. (2010) for the region around Australia…shown in Figure 7".*

**Page 14, Lines 311-313 explicit says that "ODPs for VSLSs are calculated by means of combining two sources of information: one corresponding to the slow stratospheric branch and the other to the fast tropospheric branch of transport". First, how the tropopause location is determined in the study? Second, is it possible to quantify the contribution of these two branches, and could this be taken as an approximation of the tropospheric and stratospheric influence of CHBr3 farming emissions? Note that one of the main results of the paper is that ODP-weigthed CHBr3 contribution from Aspargopsis farming would be, at most, less than 1% of the natural CHBr3 value (i.e., the Ziska_Coast scenario); thus properly showing how the ODP values were computed for this particular VSLS should be clear.**

*A: In Pisso et al. (2010), the tropopause is the WMO thermal tropopause was applied. The contributions of the two branches could be quantified but cannot be used as the approximation of the impact on troposphere and stratosphere. While the transport pathways in the troposphere and stratosphere are included, the ODPs are only calculated for the stratosphere chemistry (see also the response to comment 1a).*

**Minor Comments:**
**GENERAL: The number of significant digits used when reporting numbers should be revised throughout the hole text.**

*A: The significant figures all through the manuscript were revised. Rounded values with at most two significant figures are used for description parts, except for specific calculations (e.g. Table 1).*

**P2,L36: What do you mean by "the remains are relatively small"?**

*A: Sorry for the confusion, the sentence has been revised as "The impact of remaining farming scenarios is also relatively small".*

**P2,L37: "less than 0.016%" … is this significant different to less than 0.02%?**

*A: The value is revised as 0.02%, also see the response to the number of significances.*

**P2,L39: "by 0.48%" … of its initial value, or up to 0.48%?**

*A: revised as "up to ~0.5%"*

**P4,L88: "In consequence, the environmental impact of CHBr3 … needs to be explored and elucidated". As detailed in the main comment, the authors should explicit mention that VSLS influence both the troposphere and stratosphere, and that here only the stratospheric impact is considered.**

*A: The corresponding part in the introduction is revised as "Once released into the atmosphere, degraded halogenated VSLSs could catalytically destroy ozone both in the troposphere and*

*stratosphere, thus drawing them… explored and elucidated. In this study, only the impact on the stratosphere is considered."*

**P4,L94: I found the paper very informative not only to industry, but also to policy makers and the scientific community.**
*A: Thanks for the reviewer's recognition. The sentence is revised as "Specific objectives were to inform the industry, policy makers, as well as the scientific community on:…"*

**P5,L119: 3.4674 x 10⁴. Does this number have 5 significant digits? Please clarify and make it consistent throughout the text.**
A: *The significant figures all through the manuscript were revised. Rounded values with at most two significant figures are used for description parts, except for specific calculations (e.g. Table 1).*

**P5,L128-130: How did you get the 30 times scaling factor to extrapolate from Australian Aspargopsis production to Global production? And how did you get the 1 Tg DW value? (I could not get that value by multiplying the informed data … I must have missed something).**
*A: As described in the assumption, the Australian feedlot and dairy industries that adopted Aspargopsis is approximately equivalent to 1% of the global feedlot and dairy herds (assumption iii), and 30% of the global feed base would adopt Aspargopsis (assumption i), which results in a scaling factor of 30. As the current annual yield is 34674 Mg DW, the scaled yield to the global scale will be 34674*30 = 1040220 Mg (~1 Tg).*

**Figure 1: The lat,lon region shown in the Figure is smaller than the rectangle used for computing the average of CHBr3 mixing ratio in Figs. 4 and 6.**
*A: The larger rectangle in Figure 1 is for the convenience of better showing the locations of the farms on the map, otherwise, the locations would be almost at the edge of the plot.*

**P10,L233: Considering extending the subsection title so it includes the description of the different scenarios. In addition, by looking at Table 1 it is evident that the study was performed for meteorological conditions of year 2018 … But I could not find where in the text this is described (I might have missed it).**
*A: The L246 is revised as "We conduct FLEXPART simulations for year 2018 with different emission scenarios as…" to highlight that the study performed for 2018. The subsection title is changed to "2.4 Emission Scenarios for FLEXPART Simulations".*

**P11,L263 and Table 1: The total CHBr3 emission within the background scenario considers the well-established Ziska emission inventory, and is mentioned to consider "all 1 x1 grid cells directly neighboring the coastline", which accounts for 3109 Mg (Table 1). How large are the Ziska emissions for a small region of the size of area of Geraldton, Triabunna or Yamba? Similarly, how large are the Ziska_coast emissions if they are compared to the total Ziska emission on the Australian domain [10 -45 S, 105 -165 E] if both coastal and open-ocean grid-cells are considered?**
*A: The Ziska emissions on the domain for coastal and open ocean (also with shelf) are 3109 Mg and 2047 Mg, respectively. It is not reasonable to compute the Ziska emission on the locations of farming as some farms are terrestrial. However, if we assume all the farms are Geraldton-like (i.e.*

*all grown in the open ocean), the Ziska emission in Geraldton, Yamba and Triabunna will be 843 Mg, 295 Mg, and 676 Mg, respectively. To address this comment, these numbers due to Ziska emission are added to Section 2.4 with the sentence "Note that it is not reasonable to compute the Ziska emission on the locations of farming as some farms are terrestrial. However, if we assume all the farms are Geraldton-like (i.e., all grown in the open ocean), the Ziska emission in Geraldton, Yamba and Triabunna will be 843 Mg, 295 Mg, and 676 Mg, respectively."*

**P14,L311: The 20 days lifetime of the VSLS species considered in Pisso et al., (2010), should be mentioned here.**
*A: The sentence is revised as "…VSLS with a lifetime of 20 days, which is very similar to that…"*

**P15,L321-323: "In this study, we present the ODP-weighted emissions, which combine the information of the ODP and surface emissions and are calculated by multiplying the CHBr3 emissions with the trajectory-derived ODP at each grid point". Does Pisso et al., (2010) provide independent ODP values for each model grid-point and individual trajectory? Please see my main comment regarding this point.**
*A: The ODP values from Pisso et al. (2010) are calculated for each grid on the emission map. The values for each grid are given instead for every trajectory.*

**P15,L344 and Fig. 3a: The figure is fine, and is clear that the annual emission for the different growth periods are equivalent, but the text seems to imply that this is a new result of the study. However, these equivalent values is just a confirmation of the assumed condition that all farming scenarios for Australia must have the same total emission. This should be clarified in the text.**
*A: To clarify this, we replaced "reveal" with "show", and "indicating" is replaced with "confirming" to reinforce that this part is a conformation.*

**P16,L363: "which leads to emissions of 27 Mg (0.1 Mmol) CHBr3 per year for the targeted final yield". How do you relate this 27 Mg CHBr3 per year with the aprox. 9 Mg CHBr3 annual emission derived from Fig. 3a? Shouldn't this values be identical? Is it needed to multiply by the bromine atomicity of bromoform (3)?. Please make it clear.**
*A: The emission 9Mg from Fig 3a is based on 11558 Mg DW, which is the total annual yield of one farm. For three farm locations: Geraldton, Yamba, and Triabunna, the total annual emission will be 9Mg *3 =27 Mg.*

**Figure 3 caption: "… under different growth rates and similar initialbiomass and growth period". Please make the caption as informative as possible.**
*A: The caption is revised as "Figure 3. The annual release of $CHBr_3$ ($Mg\ yr^{-1}$) from: a) same growth rate (5%) for different growth periods; and b) under different growth rates but with same initial biomass, both a) and b) are obtained with a total annual yield of 11558 Mg DW."*

**P19,L430: "and signals with comparable magnitudes are found at 15 km". The magnitudes are comparable, but I expect this signals affect much smaller regions due to the localized source. Is this the case? If so, please make it explicit for the reader.**
*A: The sentence is revised as "…and signals with comparable magnitudes, though with smaller coverage, are found at 15 km."*

**Figs. 4 and 5: Is the color scale maximum value correct? i.e. 0.05 ppt for Fig. 4 and 0.10 ppt for Fig. 5? How large are the maximum values within the MBL? I would expect them to be much larger than the maximum value of the scale.**
*A: The difference of the color scales between Fig 4 and 5 is due to the averaging over the domain, which leads to smaller mean values. Also, for the convenience of comparing the signals due to each emission scenario, especially in the free troposphere and stratosphere, we chose smaller color scale in Fig 4. The maximum value due to Ziska_Coast in the MBL is ~0.15 ppt.*

**The caption of Fig. 4 should also explicitly indicate that it refers to Global scenarios.**
*A: The caption for Figure 4 is revised as "Altitude-time cross-sections of $CHBr_3$ mixing ratio averaged over [10°-45° S, 105°-165° E] from Global Scenarios: a) GTY_O60_30x, b) GTY_O96_Jan_30x, c) GTY_O96_Jul_30x, d) Darwin_O60_30x, and Background Scenario: e) Ziska_Coast."*

**Figure 7: If the units of the scale is a relative value between 0 and 1, please make it explicit.**
*A: The caption of Figure 7 is revised as "Spatial distribution of the ODP in January and July from Pisso et al. (2010), plotted with interval of 0.01"*

**Figure 8: The bottom-most bar presenting values for the Global Emission, for which of the global scenarios apply?**
*A: The Global Emission is taken from Tegtmeier et al. (2015) as a reference number, not from the emission scenarios in this study. To avoid the confusion, the caption of Fig 8 is revised as "Figure 8. The ODP-weighted emissions of $CHBr_3$ for Global Scenarios (GTY_O60_30x and Darwin_O60_30x), Australian Scenarios (GTY_O60 and Darwin_O60), Coastal Australian emission (Ziska_Coast), and global ODP-weighted emission for 2005 taken from Tegtmeier et al. (2015) as a reference, note that the x-axis is exponential."*

**Language editing comments:**
**GENERAL: A language style revision should be performed to the whole text**
**(as well as figure captions), mainly on the unification of past, present and future terms (is, was, will) into a common verbal tense.**
*A: We've revised the verbal tense all through the manuscript.*

**P2,L26-30: Split the sentence.**
*A: The sentence is split as "In this study, we focus on the impact of $CHBr_3$ on the stratospheric ozone layer resulting from potential emissions from proposed Asparagopsis cultivation in Australia. The impact is assessed by weighting the emissions of $CHBr_3$ with its ozone depletion potential (ODP), which is traditionally defined for long-lived halocarbons but has been also applied to very short-lived substances (VSLSs)."*

**P2,L30: DW acronym is not used again in the abstract.**
*A: The acronym DW is removed.*

**P3,L48: Two blank spaces.**
*A: revised*

**P3,L64: rephrase "showed the most potential for CH4 production decrease".**
*A: The sentence is revised as "...showed the most potential for reducing $CH_4$ production..."*

**P9,L189: What do you mean by "as the farming aims at high yielding CHBr3 varieties"?**
*A: The algae in our study are varieties with higher CHBr3 content and yield than those in the wild. To avoid confusion, the sentence is revised as "These content and release rates are higher than those for wild stock algae (Leedham et al., 2013; Nightingale et al., 1995) as the farming aims at algae varieties with high $CHBr_3$ yield."*

**P10,L214: "the gradient is between" … it is computed between? It is computed considering …?**
*A: revised as "The gradient is computed between..."*

**P15, L323-324: "The ODP-weighted emissions provide insight in where and when CHBr3 is emitted that impacts stratospheric ozone (Tegtmeier et al., 2015)". Not sure if the sentence is properly written. Please rephrase.**
*A: The sentence is rephrased as "The ODP-weighted emissions provide insight into key factors of $CHBr_3$ emission (i.e. where and when $CHBr_3$ is emitted) that impact stratospheric ozone (Tegtmeier et al., 2015)."*

**P15,L329-330: "but has no large impact on the here presented comparison of global ODP-weighted CHBr3 emissions with farm-based ODP-weighted CHBr3 emissions.". Please rephrase.**
*A: The sentence is rephrased as "but has no large impact on the comparison between global ODP-weighted $CHBr_3$ emissions and farm-based ODP-weighted $CHBr_3$ emissions presented here."*

**P19,L423: The authors use the terms "destroy" to refer to the impact of cyclone Joyce on the Australian coast. Please consider using a different wording (here and elsewhere).**
*A: The term "destroy" is replaced by "farm could be totally damaged by cyclone Joyce"*

**P23,L488: remove "again"**
*A: removed*

**P25,L517,520: (here and elsewhere). Use subindex for 3 in CHBr₃.**
*A: revised*

**P26,L525: "The local CHBr3 emissions from the Asparagopsis farms could be larger than emissions from coastal Australia." The term "local" here is correct, but seems hidden in the sentence and could be reinforced.**
*A: The sentence is revised as "The $CHBr_3$ emissions from the localized Asparagopsis farms".*

References:

Claxton, T., R. Hossaini, O. Wild, M.P. Chipperfield, and C. Wilson, On the regional and seasonal ozone depletion potential of chlorinated very shortlived substances, Geophys. Res. Lett., 46(10), 5489–5498. doi:10.1029/2018GL081455, 2019.

Fernandez, R.P., J.A. Barrera, A.I. L pez-Nore a, D.E. Kinnison, J. Nicely, R.J. Salawitch, P.A. Wales, B.M. Toselli, S. Tilmes, J.-F. Lamarque, C.A. Cuevas, and A. Saiz-Lopez, Intercomparison between surrogate, explicit and full treatments of VSL bromine chemistry within the CAM-Chem chemistry-climate model, Geophys. Res. Lett., 48(4), doi:10.1029/2020GL091125, 2021.

Pisso, I., P.H. Haynes and K. S. Law, Emission location dependent ozone depletion potentials for very short-lived halogenated species, Atmos. Chem. Phys., 10, 12025–12036, https://doi.org/10.5194/acp-10-12025-2010, 2010.

Zhang, J., D.J. Wuebbles, D.E. Kinnison, and A. Saiz-Lopez, Revising the ozone depletion potentials metric for short-lived chemicals such as CF3I and CH3I. J. Geophys. Res. Atmos., 125, e2020JD032414. https://doi.org/10.1029/2020JD032414, 2020.

---

## Author Comment (AC3)

Dear Reviewers

We would like to express our sincere gratitude to the reviewers for your effort to improve our manuscript. Based on your comments, we've revised our manuscript accordingly with changed parts marked red.

The following is the point-to-point response with reviewers' comments in bold and the responses italic.

**Comment on acp-2021-800**
**Anonymous Referee #3**
**Referee comment on "Potential environmental impact of bromoform from Asparagopsis farming in Australia" by Yue Jia et al., Atmos. Chem. Phys. Discuss.,**
**https://doi.org/10.5194/acp-2021-800-RC3, 2022**
**Jia et al. presented a modeling analysis on the potential environmental impact of bromoform from Australia Asparagopsis farming on atmospheric ozone depletion. This is an interesting environment impact analysis, and the results would be of great interest to the Asparagopsis farming community and some environmental policy makers in Australia. Overall, the experiment is adequately designed, and the paper is well written, and should be accepted for publication on ACP. I have only some very minor comments. Change "long-lived halogens" to either long-lived halogen-containing compounds or long-lived halocarbons. ODP is not defined for long-lived halogens.**
*A: The phrase has been changed to "long-lived halocarbons".*

**->varies, depending on**
*A: revised*

**-> emitted into the atmosphere**
*A: revised*

**Here and later in the text: To cite the WMO assessment chapters, you should use "Engel and Rigby et al, 2018"**
*A: corrected*

**I would recommend rephrasing of "the aim of this study was elucidation of …" to "… to assess the impact/contribution …". Elucidation seems to be too much of an assertion in the context of this paper.**
*A: revised*

**an effective**
*A: corrected*

**L305-307. Is there a published reference for trajectory-based ODP? Brioude et al. (2010) and Pisso et al. (2010)??**
*A: The references have been added.*

**L326 & L329. May be this is a personal habit thing, but I would prefer "the OPD values applied here" and "the comparison presented here"**
*A: revised.*

**L520, 527, 532, 542, CHBr"3" should be subscript.**
*A: The subscript all through the manuscript are corrected*

---

## Author Response (AR2)

Dear Editor

Thank you for your comments to improve our manuscript. According to your suggestion, we've revised our manuscript. The changed parts in the manuscript are marked red. The following is a point-to-point response.

Best,
Yue on behalf of co-authors.

**Comments to the author:**
**Dear authors,**

**Thank you for addressing the reviewer comments. I have one remaining issue I would like you to look at. In the revised manuscript you mention (line 569) that "... this would not really increase ". This sounds a little unscientific to me. Please give a value, so the magnitude of the effect is clear.**

**Andreas Engel**

A: Thanks for pointing this out, indeed the sentence sounds a little unscientific. The flux in Ziska emission is calculated as $F = k_w \cdot \Delta c = k_w \cdot (c_w - \frac{c_{atm}}{H})$ (Eq. 3). As $k_w$ is a function of wind speed and SST, the flux is determined by local meteorology and the air-sea concentration gradient ($c_w - \frac{c_{atm}}{H}$). For new flux calculations simultaneous measurements in water and air are required, which are currently not available. For our example if only the atmospheric $CHBr_3$ abundance $c_{atm}$ increases, the corresponding flux will decrease. In particular, the $CHBr_3$ mixing ratio near Cape Grim used to calculate Ziska emissions is ~0.8 ppt, with a corresponding flux 109 pmol m$^{-2}$ h$^{-1}$. If $c_{atm}$ is increased to 1.5 ppt, the corresponding flux will be reduced to a flux from the atmosphere into the ocean of -113 pmol m$^{-2}$h$^{-1}$. However, it is clear that in order to maintain 1.5 ppt atmospheric mixing ratio, high air-sea fluxes (driven by high oceanic concentrations) would be required. Given that we don't know the size of updated oceanic concentrations, we are not able to

provide a new air-sea flux value, but have instead added the following text to explain the situation: 'New $CHBr_3$ measurements in Cape Grim close to Triabunna show larger $CHBr_3$ mixing ratios (~1.5 ppt, Dunse et al., 2020) than the Ziska climatology (~0.8 ppt, Ziska et al., 2013). Similarly, the Ziska climatology is known to underestimate water concentrations of $CHBr_3$ in coastal regions with spare local measurements (Ziska et al., 2013; Maas et al., 2021). While the new atmospheric measurements suggest that a higher flux is required than currently included in the Ziska climatology, updated air-sea flux values can only be derived for simultaneous measurements in water and air, which are currently not available. It is important to note that such updated air-sea flux estimates would only impact the conclusions of our study if they would be much lower than the old estimates over large parts of the Australian coastline, a scenario which is highly unlikely.'